# 3D Equivariant Pose Regression via Direct Wigner-D Harmonics Prediction

**Jongmin Lee**[*]     **Minsu Cho**

Pohang University of Science and Technology (POSTECH), South Korea
{ljm1121, mscho}@postech.ac.kr
http://cvlab.postech.ac.kr/research/3D_EquiPose

## Abstract

Determining the 3D orientations of an object in an image, known as single-image pose estimation, is a crucial task in 3D vision applications. Existing methods typically learn 3D rotations parametrized in the spatial domain using Euler angles or quaternions, but these representations often introduce discontinuities and singularities. SO(3)-equivariant networks enable the structured capture of pose patterns with data-efficient learning, but the parametrizations in spatial domain are incompatible with their architecture, particularly spherical CNNs, which operate in the frequency domain to enhance computational efficiency. To overcome these issues, we propose a frequency-domain approach that directly predicts Wigner-D coefficients for 3D rotation regression, aligning with the operations of spherical CNNs. Our SO(3)-equivariant pose harmonics predictor overcomes the limitations of spatial parameterizations, ensuring consistent pose estimation under arbitrary rotations. Trained with a frequency-domain regression loss, our method achieves state-of-the-art results on benchmarks such as ModelNet10-SO(3) and PASCAL3D+, with significant improvements in accuracy, robustness, and data efficiency.

## 1 Introduction

Predicting the 3D pose of objects, i.e., position and orientation, in 3D space from an image is crucial for numerous applications, including augmented reality [59], robotics [4, 5, 63, 69], autonomous vehicles [21, 50], and cryo-electron microscopy [79]. Estimating 3D orientation is particularly challenging due to rotational symmetries and the non-linear nature of rotations. In addition, unlike translations, rotations introduce unique challenges such as gimbal lock and the requirement for continuous, singularity-free representations. Existing methods often learn 3D rotations using spatial domain parameterizations like Euler angles, quaternions, or axis-angle representations, as illustrated in Figure 1. However, these parameterizations suffer from issues such as discontinuities and singularities [54, 58, 80], which can hinder the performance and reliability.

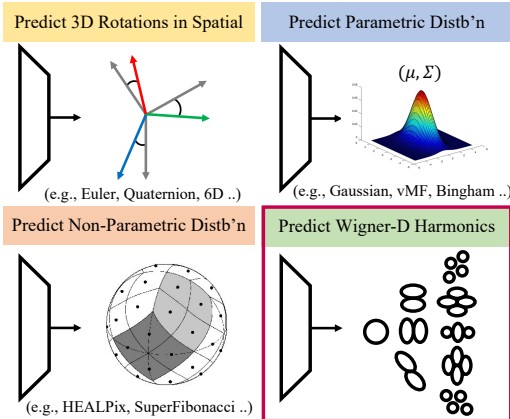

Figure 1: **Types of representations for 3D rotation prediction.** Existing methods consider predicting 3D rotations in the spatial domain. Our method predicts Wigner-D coefficients in the frequency domain, to obtain accurate pose in continuous space using an SO(3)-equivariant network.

---

[*]The current affiliation of Jongmin Lee is with LG AI Research. Contact: jongminlee@lgresearch.ai.

38th Conference on Neural Information Processing Systems (NeurIPS 2024).

SO(3)-equivariance enables accurate 3D pose estimation and improves generalization to unseen rotations. It ensures that outputs consistently change with the 3D rotation of the input, maintaining rotational consistency between the input and output across network layers. Despite its importance, many existing methods [3, 44, 52, 75, 80] often design networks without considering SO(3)-equivariance, resulting in suboptimal performance when dealing with 3D rotations. In addition, in the context of spherical CNNs [9, 10, 12, 16–18, 36] for efficient SO(3)-equivariant operations, the 3D rotation parametrization in the spatial domain is inadequate because these SO(3)-equivariant networks operate in the frequency domain.

To address these challenges, we propose an SO(3)-equivariant pose harmonics regression network that directly predicts Wigner-D coefficients in the frequency domain for 3D rotation regression. Building on prior work [28, 35], our method leverages the properties of spherical CNNs [11], which operate in the frequency domain, to guarantee SO(3)-equivariant output representation. By directly regressing Wigner-D matrix coefficients, our approach eliminates the need to convert outputs into spatial representations during training, ensuring alignment with the operations of spherical CNNs. This design allows us to bypass the limitations inherent in traditional spatial parameterizations—such as discontinuities and singularities [54, 58, 80]—resulting in more precise and continuous pose estimation. We further introduce a frequency-domain MSE loss to enable continuous training of 3D rotations, with the flexibility to incorporate distributional losses [52] for effectively capturing rotational symmetries in objects. Our method achieves state-of-the-art performance on standard single object pose estimation benchmarks, including ModelNet10-SO(3) and PASCAL3D+, demonstrating high sampling efficiency and strong generalization to unseen 3D rotations.

## 2 Related Work

**SO(3) pose regression.** The choice of rotation representation is a fundamental aspect of the current SO(3) pose estimation methods. In the early stages of deep learning, methods for SO(3) pose regression choose the rotation representation by direct cosine matrices [29, 73], Euler angles [37, 48, 49, 60, 61], quaternions [5, 13, 31, 32, 70, 78], and axis-angles [14, 20, 62]. However, according to [80], for any representation $R$ in a Euclidean space of dimension $d \leq 4$, such as Euler angles and quaternions, $R$ is discontinuous and unsuitable for deep learning. In addition, Euler angles can cause gimbal lock, which restricts certain rotations, whereas quaternions avoid this issue but their double representation of rotations in SO(3) can lead to complications such as local minima in optimization problems. As an alternative, a continuous 6D representation with Gram-Schmidt orthonormalization [80] and 9D representation with singular value decomposition (SVD) [8, 41] have been proposed, and [7] proposes manifold-aware gradient layer to facilitate the learning of rotation regression. Denoising diffusion models are employed in the context of SO(3) pose regression [64], or for solving pose estimation by aggregating rays [76]. In contrast to existing SO(3) pose regression methods that formulate rotation representations in the spatial domain, we define the Wigner-D coefficients as the output of the network in the frequency domain, using SO(3)-equivariant networks.

**Pose estimation with a parametric distribution.** To model rotation uncertainty, parametric distributions on the rotation manifold are employed in a probabilistic manner. [57] predicts parameters of a mixture of von Mises distributions over Euler angles using Biternion networks. [5, 13, 23] utilize the Bingham distribution over unit quaternions to generate multiple hypotheses of rotations. [51, 74] leverage the matrix Fisher distribution [33] to construct a probabilistic model for SO(3) pose estimation. Additionally, [75] propose the Rotation Laplace distribution for rotation matrices on SO(3) to suppress outliers, and the Quaternion Laplace distribution for quaternions on $\mathcal{S}^3$. Nevertheless, parametric models rely on predefined priors. In contrast, our model uses non-parametric modeling during inference to capture more complex pose distributions.

**Pose estimation with a non-parametric distribution.** Probabilistic pose estimation can also be achieved by predicting non-parametric distributions. IPDF [52] introduces the estimation of arbitrary, non-parametric distributions on SO(3) using implicit functions with MLPs, and Hyper-PosePDF [27] uses hypernetworks to predict implicit neural representations by Fourier embedding. ExtremeRotation [6] predicts discretized distributions over $N$ bins for relative 3D rotations trained with cross-entropy loss. RelPose [43, 77] uses an energy-based formulation to represent distributions over the discretized space of SO(3) relative rotation. Several SO(3)-equivariant modeling methods construct non-parametric distributions by utilizing icosahedral group convolution [34], projecting image features orthographically onto a sphere [35], and satisfying consistency properties of SO(3)

by translating them into an SO(2)-equivariance constraint [28]. RotationNormFlow [44] uses discrete normalizing flows to directly generate rotation distributions on SO(3). These non-parametric methods, which are trained with loss functions in discretized distributions, such as cross-entropy and negative log-likelihood, tend to lose precision in rotation prediction. In contrast, our method predicts continuous SO(3) transformations through regression, eliminating the need to approximate SO(3) poses within a discretized space and enabling our model to achieve accurate 3D rotations.

## 3 Preliminary

### 3.1 Representations of Rotations

**Rotation representation in spatial domain.** In 3D rotation, Euler angles are a common SO(3) representation but suffer from non-uniqueness and gimbal lock, making them less suitable for neural network predictions. Quaternions offer a solution by preventing gimbal lock, but their non-unique representation (q and -q) can complicate certain optimization processes. The axis-angle representation is intuitive but can encounter singularities. The 6D and 9D representations provide newer approaches that simplify optimization in deep networks by avoiding non-linear constraints and ensuring orthogonality. However, they also introduce complexities in maintaining constraints during the learning process. Thus, choosing an appropriate rotation representation is crucial for accurate pose estimation in various computational applications. For a detailed explanation, please refer to Sec. A.1 and an overview of learning 3D rotations in [54, 58].

**Rotation representation in frequency domain.** In the frequency domain, 3D rotation is managed by manipulating spherical harmonics coefficients. Spherical harmonics, denoted as $Y_m^l(\theta, \phi)$, are functions defined on the surface of a sphere using polar ($\theta$) and azimuthal ($\phi$) angles. These harmonics are characterized by their degree $l$ and order $m$, truncated to a maximum degree $L$ for computational feasibility. The rotation of spherical harmonics is represented by the shift theorem [46], where a rotation operator $\Lambda_g$ acts on spherical harmonics, transforming them via a matrix $D_{mn}^l(g)$:

$$\Lambda_g Y_m^l(x) = \sum_{|n| \leq l} D_{mn}^l(g) Y_n^l(x). \tag{1}$$

This matrix, part of the irreducible unitary representation of SO(3), expresses how each harmonic changes under rotation, summing over all orders $n$ from $-l$ to $l$, called Wigner-D matrix. The Wigner-D rotation representation is not limited to a specific case of 3D rotations but can be converted from any 3D rotation representation, such as Euler angles, quaternions, and 3D rotation matrices. Our SO(3) equivariant network predicts the Wigner-D representation in the frequency domain instead of predicting rotations in the spatial domain. For a detailed explanation, please refer to Sec. A.2.

### 3.2 SO(3)-Equivariance

**Equivariance.** Equivariance is a useful property to have because transformations $T$ applied to the input produce predictable and consistent output of the features through transformations $\phi \in \Phi$, enhancing both interpretability and data efficiency. For example, a feature extractor $\Phi$ is equivariant to a transformation if applying the transformation to the input and then applying the extractor produces the same output as applying the extractor first and then the transformation:

$$\Phi(T_g(x)) = T_g'(\Phi(x)) \tag{2}$$

where $T_g$ and $T_g'$ represent transformations acting on a group $g \in G$ of the input and output spaces, respectively. This ensures that the network's output remains consistent with transformations applied to the input. For translation groups, convolution inherently maintains this property. For rotations, additional rotation-equivariant layers are integrated into the network design.

Group-equivariant convolutional networks [10] extend this concept to complex groups like rotations or other symmetries. By designing convolutions that are equivariant to these group actions, these networks can handle a broader range of transformations. This can be mathematically described as:

$$[h * \phi](g) = \sum_{y \in X} h(y) \cdot \phi(g^{-1}y) \tag{3}$$

where $h$ is the input function over space $X$, $\phi$ is the filter or kernel, and $g \in G$ is an element of the group. The term $g^{-1}y$ represents the transformation of $y$ by the inverse of $g$. This operation

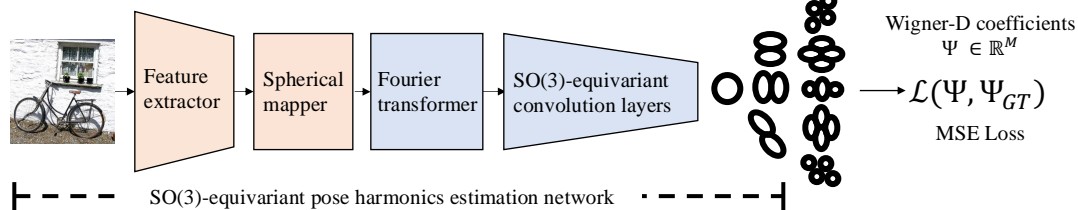

**Figure 2: Overall architecture.** Our network for SO(3)-equivariant pose estimation consists of four parts: feature extraction, spherical mapper, Fourier transformer, and SO(3)-equivariant layers. First, we extract a feature map using a pre-trained ResNet. Next, the spherical mapper orthographically projects the extracted feature map onto a spherical surface. The Fourier transformer converts this spatial information into the frequency domain. We utilize spherical convolutions to obtain the final Wigner-D harmonics coefficients $\Psi$ which represent SO(3) rotations of spherical harmonics, where $M$ denotes the total number of Wigner-D matrix coefficients.

ensures that the network remains equivariant to the actions of the group $G$, allowing it to handle inputs transformed by any element of this symmetry group.

On the sphere in 3D, however, there is no straightforward way to implement a convolution in the spatial domain due to non-uniform samplings [12]. This challenge arises because traditional convolution operations rely on uniform grid structures, which are not applicable to spherical data. To address this, specialized methods such as spherical convolutions or graph-based approaches are employed to handle the unique structure and sampling patterns of spherical data, thereby ensuring effective feature extraction and equivariance on spherical surfaces.

**Spherical convolutions for $SO(3)$-equivariance.** To effectively analyze complex spatial data, such as for volumetric rendering and 3D pose estimation, it is necessary to develop functions with equivariance to the $SO(3)$ group. Early methods for spherical convolution were defined by computing Fourier transforms and convolution on the 2-sphere [15]. However, the output of these spherical convolutions is a function on the sphere, not on $SO(3)$. Spherical CNNs [11] extended this approach to effectively convolve on the $SO(3)$ group. Using the truncated Fourier transform, signals on $S^2$ are modeled with spherical harmonics $Y_n^l$, and on $SO(3)$ with Wigner-D matrix coefficients $D_{mn}^l$.

To efficiently compute the $S^2$ and $SO(3)$ convolution, generalized fast Fourier transforms (GFFTs) demonstrate optimized computation [11]. The GFFTs show robustness and efficiency in spherical signal processing, where the spectral group convolutions become simpler element-wise multiplications in the Fourier domain. Specifically, for $S^2$, the process uses vectors of spherical harmonic coefficients, forming a block diagonal matrix analogous to $SO(3)$ convolution. Both convolutions on $S^2$ and $SO(3)$ generate output signals that reside on $SO(3)$.

## 4 SO(3)-Equivariant Pose Harmonics Predictor

The goal of our network is to accurately predict the SO(3) pose of an object in an image. To achieve, we employ spherical CNNs [11] to obtain SO(3)-equivariant representation, and our model is trained with frequency-domain supervision using Wigner-D coefficients. This approach enhances data efficiency by capturing patterns with fewer training samples and ensures precise SO(3) pose estimation by aligning the parametrization of 3D rotations with the Wigner-D matrices in the frequency domain.

Figure 2 provides an overview of our SO(3)-equivariant pose estimation network. In Sec. 4.1, we explain the steps for obtaining the Wigner-D representation, following the method described in [35]. In Sec. 4.2, we introduce a frequency-domain regression loss, where we train the network using MSE loss between the predicted representation and the ground truth (GT) Wigner-D coefficients. Finally, in Sec. 4.3, we describe the inference process by constructing an SO(3) grid for evaluation.

### 4.1 SO(3)-Equivariant Pose Estimation Network

In this subsection, we explain our SO(3)-equivariant pose estimation network, highlighting that its key components are shared with the architecture of [35].

**Image feature extraction.** We first apply a feature extractor to obtain an image feature map that encodes semantic and geometric information: $F = \rho(I)$, where $F \in \mathbb{R}^{C \times H' \times W'}$ and $\rho$ denotes ResNet. We utilize a ResNet feature extractor that is pre-trained on ImageNet same to [28, 35, 44, 51, 52, 75]. We then perform dimensionality reduction on the image feature $F$ to match the input dimension of the subsequent spherical feature using a 1x1 convolution: $F' = \text{Conv}_{1 \times 1}(F)$, where $F' \in \mathbb{R}^{C' \times H' \times W'}$.

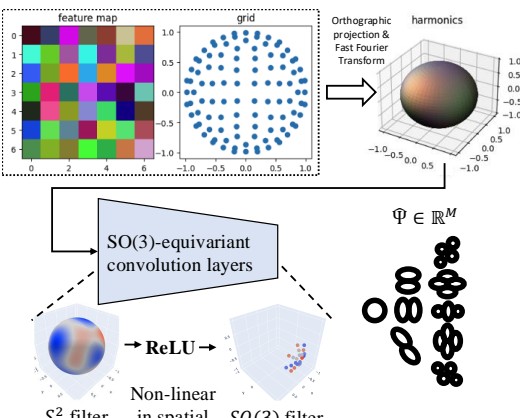

**Spherical mapper.** To begin, we lift the image features to the 2-sphere using orthographic projection [35]. This involves mapping the 2D feature $F'$ to a spherical feature $\psi \in \mathbb{R}^{C' \times p}$, where $p$ denotes the number of points on the sphere. The orthographic projection links pixels in the image space to points on the sphere by orthogonally mapping $S^2$ coordinates to the image plane, thereby preserving the spatial information of the dense 2D feature map.

Initially, we model spherical coordinates using an $S^2$ HEALPix [24] grid over a hemisphere. Within this hemisphere, the set $\{x_i\} \subset S^2$ represents the vertices of the grid. Each vertex $x_i$ is mapped to a position $P(x_i)$ on the image plane. Formally, the orthographic projection $P$ maps 3D coordinates on the hemisphere to 2D coordinates on the image plane as $P(x, y, z) = (x, y)$.

Due to the fixed perspective, only one hemisphere of the sphere is visible, resulting in a localized signal $\psi(x) = F'(P(x))$ supported over this hemisphere. The value of $\psi(x_i)$ is ob-

Figure 3: **Illustration of spherical mapper and spherical convolution for SO(3)-equivariance.** This structure allows for the prediction of 3D rotations while preserving the SO(3)-equivariance of the input structure. Predicting the Wigner-D harmonics $\Psi$ enables continuous 3D rotation modeling, without discretizing the group actions.[1]

tained by interpolating $F'$ at the pixels near $P(x_i)$ in the image space using an interpolation function $\eta$, so $\psi(x_i) = \eta(F', P(x_i))$. Figure 3 illustrates the processes of the spherical mapper, and the following frequency domain conversion and spherical convolution for SO(3)-equivariance.

**Convert to the frequency domain.** The transition of the spherical feature $\psi$ into the frequency domain is achieved using the fast Fourier transform (FFT) adapted for spherical topology. By employing the FFT, we efficiently convert $\psi$ to spherical signals $\mathcal{S}$, represented as a sum of spherical harmonics. This transformation allows us to capture and manipulate the spatial frequencies inherent to the spherical surface. Specifically, the transition to the frequency domain enables the derivation of Wigner-D coefficients, which effectively model the $SO(3)$. The Fourier series of $\mathcal{S}$ is truncated at frequency $L$ [11], expressed as: $\mathcal{S}(x) \approx \sum_{l=0}^{L} \sum_{m=-l}^{l} c_m^l Y_m^l(x)$, where $\mathcal{S} \in \mathbb{R}^{C' \times N}$, $N$ is the total number of spherical harmonics determined by the maximum frequency $L$, and $Y_m^l(x)$ are the spherical harmonics. Operating in the frequency domain facilitates the effective convolution of signals on the sphere $(S^2)$ and within the 3D rotation group $(SO(3))$, preserving the geometric properties of input features through spherical equivariance.

To address sampling errors from approximating the Fourier series via truncation, we apply two techniques proposed in [35]. First, to prevent discontinuities on the 2-sphere, we gradually decrease the magnitude of projected features near the image edge: $\psi'(x_i) = w(x_i) \cdot \psi(x_i)$. Second, for each projection, we randomly select a subset of grid points on the $S^2$ HEALPix grid as a dropout.

**Spherical convolution for SO(3)-equivariance.** We aim to predict 3D rotations while preserving SO(3)-equivariance using the projected features on sphere. First, the spherical signal $\mathcal{S}$ is processed with an $S^2$-equivariant convolutional layer [11, 35]. Unlike conventional convolutions with local filters, $S^2$ convolution uses globally supported filters, offering a global receptive field. This allows for a shallower network, which is important due to the high computational and memory demands of spherical convolutions at a high bandlimit $L$.

---

[1]We use the visualization tools available in the source code of the image2sphere GitHub repository.

In this stage, we obtain SO(3) representations inherent to spherical CNNs [11]. The output of $S^2$ convolutions lies in the SO(3) domain because $S^2$ convolutions replace translations with rotations, and the space of 3D rotations forms the SO(3) group. Consequently, we obtain feature results sized in $\mathbb{R}^{C'' \times M}$, where $C''$ is the hidden dimension of the SO(3) features, and $M$ is the total number of Wigner-D matrix coefficients, given by $M = \sum_{l=0}^{L}(2l + 1) \times (2l + 1)$, created by SO(3) irreps.

We apply non-linearities between convolutional layers by transforming the signal to the spatial domain, applying a ReLU, and then transforming back to the frequency domain, following the approach of spherical CNNs [11]. This method can be extended to FFT-based approximate non-linearity [19] and equivariant non-linearity for tensor field networks [55, 71].

Subsequent to the $S^2$-equivariant convolutional layer, we perform an SO(3)-equivariant group convolution [11, 35] using a locally supported filter to refine the SO(3) pose space. Unlike typical spherical CNNs, we bypass the inverse fast Fourier transform (iFFT) and instead use the output harmonics of Wigner-D prediction. This approach, unlike that of [35], improves the efficiency of our method. The final output of the equivariant network is the Wigner-D matrix coefficients $\Psi \in \mathbb{R}^M$.

## 4.2 Frequency-Domain Regression Loss

The output of the SO(3)-equivariant convolutional layers is a linear combination of Wigner-D matrices, represented as a flattened vector of the Wigner-D coefficients. The output $\Psi$ indicates specific object orientations in an image. To generate the ground-truth (GT) Wigner-D coefficients, we convert the GT 3D rotations from Euler angles using the $ZYZ$ sequence of rotation $R$, expressed as $R = R_z(\gamma)R_y(\beta)R_z(\alpha)$ to the Wigner-D matrices $D_{mn}^l(\alpha, \beta, \gamma)$, where $D$ represents an action of the rotation group SO(3). We calculate the Mean Squared Error (MSE) loss as follows:

$$\mathcal{L}(\hat{\Psi}, \Psi^{\mathrm{GT}}) = \sum_{l=0}^{L} \sum_{m=-l}^{l} w_l(\hat{\Psi}_{lm} - \Psi_{lm}^{\mathrm{GT}})^2, \qquad (4)$$

where $w_l$ are weights assigned to each harmonic frequency level $l$, normalizing the output Wigner-D matrices for a frequency-domain specific MSE loss. This loss function enables continuous prediction of SO(3) poses using SO(3)-equivariant networks, whereas the previous methods [28, 35, 52] predicted outputs in a discretized distribution, leading to degradation in prediction precision. With this re-parametrization in the frequency domain, we use Euclidean distance because it is simple yet effective for pose prediction. It allows straightforward calculation while considering both the direction and magnitude of the vectors. Many distance metrics defined in the spatial domain [2, 25, 30, 58] may not be directly appropriate for the frequency domain without adaptation. For example, cosine and angular distances ignore magnitude, where the amplitude of frequency components carries significant information. Chordal and geodesic distances require normalization, can be less intuitive, and often involve more complex computations.

## 4.3 Inference

For evaluation, the output Wigner-D representation $\Psi$ is converted to an SO(3) pose in the spatial domain. Figure 4 illustrates the inference process inspired by [35, 52]. Specifically, we map the predicted Wigner-D coefficients $\hat{\Psi}$ from the frequency domain to a 3x3 rotation matrix $R$ by querying $\hat{\Psi}$ on a predefined SO(3) grid. To achieve this mapping, we calculate the similarities between the output vector $\Psi$ and the SO(3) grid $P(\cdot \mid I)$. These similarities are then normalized using a softmax function to produce a non-parametric categorical distribution $P(R \mid I)$. The final 3D rotation matrix $\hat{R}$ is determined either by taking the argmax of this distribution or by applying gradient ascent [52].

To generate the SO(3) equivolumetric grids, we utilize the hierarchical equal area isolatitude pixelation of the sphere (HEALPix) [24, 26], consistent with methods used in [28, 35, 44, 52, 75]. To lift the $S^2$ HEALPix to $SO(3)$ HEALPix, we create equal-area grids on the 2-sphere and cover $SO(3)$ by threading great circles through each point using the Hopf fibration from [72].

This inference scheme effectively models objects with ambiguous orientations or symmetries by employing multiple hypotheses, thereby overcoming the limitations of single-modality predictions [47]. In addition to joint training with distributional cross-entropy loss [52], our network can model the non-parametric and multi-modal distribution in pose space to address pose ambiguity and aid in modeling 3D symmetry.

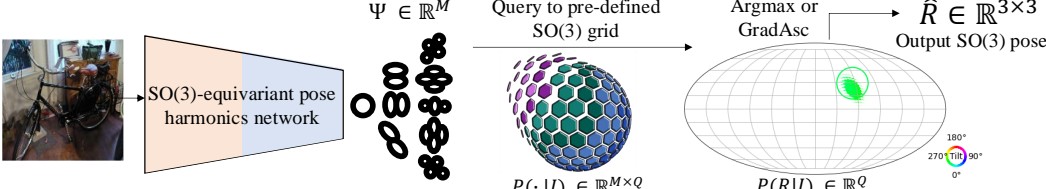

Figure 4: **Inference time.** We query the output vector of Wigner-D coefficients $\Psi$ against the predefined SO(3) HEALPix grid with a resolution of $Q$ points. We finally obtain the SO(3) probability distribution $P(R \mid I)$, where each position represents the probability of a specific SO(3) pose.[2]

# 5 Experiment

## 5.1 Implementation Details

We input a 2D RGB image $I \in \mathbb{R}^{3 \times 224 \times 224}$. A ResNet backbone, pretrained on ImageNet, extracts feature maps of shape $F \in \mathbb{R}^{2048 \times 7 \times 7}$. We then perform dimension reduction using a 1x1 convolution to obtain $F' \in \mathbb{R}^{512 \times 7 \times 7}$. In the spherical mapper, the features are mapped onto an $S^2$ grid generated by recursion level 2 of HEALPix on half of the sphere, and then sampled at 20 points, resulting in $\psi \in \mathbb{R}^{512 \times 20}$. By converting $\psi$ to the frequency domain, the spherical signals $\mathcal{S} \in \mathbb{R}^{512 \times 49}$ are obtained. The Wigner-D representation is implemented in a flattened form across different frequency levels. For example, the matrix coefficients at a frequency level $l$ are represented as a flattened vector of size $(2l + 1) \times (2l + 1)$. We use a maximum frequency level of $L = 6$, resulting in a total size of $M = 455$, computed as $\sum_{l=0}^{6}(2l + 1) \times (2l + 1)$. These coefficients are then flattened into a single vector for the Wigner-D prediction. The spherical convolution on the $S^2$ kernel uses an 8-dimensional hidden layer with global support to obtain intermediate SO(3) features in $\mathbb{R}^{8 \times 455}$. After nonlinear activation, we finally obtain the 1-dimensional output $\Psi \in \mathbb{R}^{1 \times 455}$ using an SO(3) convolution with a locally supported filter to handle rotations up to $22.5°$. At inference, we employ a recursive level 5 of SO(3) HEALPix grid with 2.36 million points, achieving a precision of $1.875°$, as in [28, 35].

## 5.2 Benchmarks

**ModelNet10-SO(3)** [42] is a common dataset for estimating a 3D rotation from a single image. The images are created by rendering CAD models from the ModelNet10 dataset [67]. The dataset includes 4,899 objects across 10 categories, each image is labelled with a single 3D rotation. The rotations are uniformly sampled from each CAD model. From a single CAD model, the training set comprises 100 3D rotations on SO(3), while the test set includes 4 unseen 3D rotations.

**PASCAL3D+** [68] is a widely-used benchmark for evaluating pose estimation in images captured in real-world settings. It includes 12 categories of everyday objects, which were created by manually aligning 3D models with their corresponding 2D images. This dataset presents challenges due to the significant variation in object appearances, the high variability of natural textures, and the presence of novel object instances in the test set. To be consistent with the baselines, we conduct training data augmentation using synthetic renderings [60].

**ModelNet10-SO(3) Few-shot Views** is used to evaluate the data efficiency of pose estimation models. Unlike the original ModelNet10-SO(3) [42], we have expanded this to evaluate various amounts of training data, by setting the number of training views per CAD model to 3, 5, 10, 20, 30, 40, 50, 70, 90, and 100. This benchmark verifies the sampling efficiency of our equivariant networks. We use the same test dataset as that of ModelNet10-SO(3).

**Evaluation Metrics.** We calculate the angular error, measured in degrees using geodesic distance, between the network-predicted SO(3) pose and the ground-truth rotation matrix: $\theta_{\text{Error}}(R, \hat{R}) = \cos^{-1}\left(\frac{\text{trace}(\Delta R) - 1}{2}\right)$, and $\Delta R = R\hat{R}^T$. We adopt two commonly used metrics: the median rotation error (MedErr) and the accuracy within specific rotation error thresholds (Acc@15° and Acc@30°).

---

[2]We use the illustration to represent the SO(3) grid $P(\cdot \mid I)$ from s2fft github repository [56].

| Method | Acc@15 | Acc@30 | Rot Err. (Median) |
|---|---|---|---|
| Zhou *et al.* [80] | 0.251 | 0.504 | 41.1° |
| Bréiger [3] | 0.257 | 0.515 | 39.9° |
| Liao *et al.* [42] | 0.357 | 0.583 | 36.5° |
| Prokudin *et al.* [57] | 0.456 | 0.528 | 49.3° |
| Deng *et al.* [13] | 0.562 | 0.694 | 32.6° |
| Mohlin *et al.* [51] | 0.693 | 0.757 | 17.1° |
| Murphy *et al.* [52] | 0.719 | 0.735 | 21.5° |
| Yin *et al.* [74] | - | 0.751 | 16.1° |
| Yin *et al.* [75] | 0.742 | 0.772 | 12.7° |
| Klee *et al.* [35] | 0.728 | 0.736 | 15.7° |
| Liu *et al.* (Uni) [44] | 0.760 | 0.774 | 14.6° |
| Liu *et al.* (Fisher) [44] | 0.744 | 0.768 | 12.2° |
| Howell *et al.* [28] | - | - | 17.8° |
| ours (ResNet-50) | 0.759 | 0.767 | 15.1° |
| ours (ResNet-101) | **0.773** | **0.780** | **11.9°** |

Table 1: **Results on ModelNet10-SO(3).** The scores have been averaged across all ten object categories.

| Method | Acc@30 | Rot Err. (Median) |
|---|---|---|
| Zhou *et al.* [80] | - | 19.2° |
| Bréiger [3] | - | 20.0° |
| Tulsiani & Malik [61] | - | 13.6° |
| Mahendran *et al.* [45] | - | 10.1° |
| Liao *et al.* [42] | 0.819 | 13.0° |
| Prokudin *et al.* [57] | 0.838 | 12.2° |
| Mohlin *et al.* [51] | 0.825 | 11.5° |
| Murphy *et al.* [52] | 0.837 | 10.3° |
| Yin *et al.* [75] | - | 9.4° |
| Klee *et al.* [35] | 0.872 | 9.8° |
| Liu *et al.* (Uni) [44] | 0.827 | 10.2° |
| Liu *et al.* (Fisher) [44] | 0.863 | 9.9° |
| Howell *et al.* [28] | - | 9.2° |
| Ours | **0.892** | **8.6°** |

Table 2: **Results on PASCAL3D+ with ResNet-101 backbone.** Scores are averaged across all twelve classes.

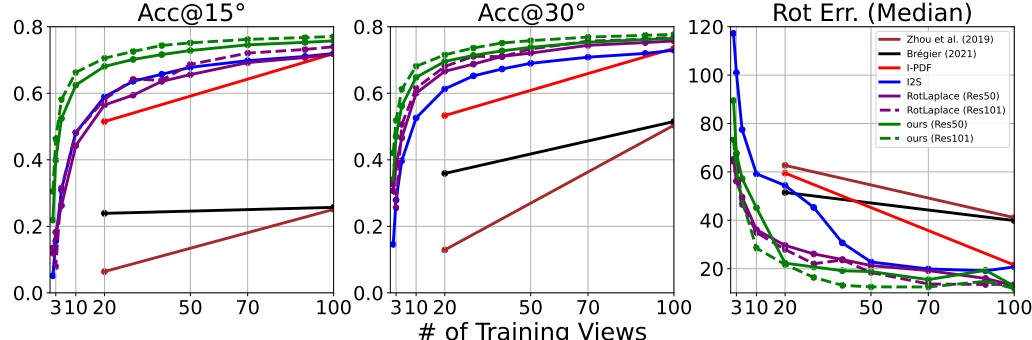

Figure 5: **Experiment on ModelNet10-SO(3) with few-shot training views.** Results with solid lines of I-PDF [52], I2S [35], and RotLaplace [75] denote to a ResNet-50 backbone, while dotted lines indicate a ResNet-101 backbone. Our method outperforms all metrics and reduces training views. Baseline results [35, 75] were obtained using the source code provided by the authors.

## 5.3 Results

Table 1 shows the pose estimation results on the ModelNet10-SO(3) dataset, where our model outperforms all baselines across multiple evaluation metrics. Notably, our Wigner-D harmonics prediction network surpasses methods in non-probabilistic rotation estimation [3, 42, 80], parametric probability distribution estimation [13, 44, 51, 57, 74, 75], and non-parametric distribution prediction [28, 35, 52], by leveraging SO(3) equivariance and rotation parametrization in the frequency domain.

Table 2 shows the results on the PASCAL3D+ benchmark. Our SO(3) Wigner-D harmonics predictor achieves state-of-the-art performance on these challenging benchmarks. It demonstrates robustness to changes in object appearance, real textures, and generalizes well to novel object instances. We additionally report fine-scale accuracies, i.e., Acc@3°, Acc@5°and Acc@10°, in Tables A1 and A2, and class-wise evaluation of the results in Tables A3 and A4 of appendix B.

## 5.4 Few-shot Training Views

Figure 5 illustrates the pose estimation results on ModelNet10-SO(3) with few-shot training views. Notably, as the number of training views from a single CAD model decreases, our model consistently achieves the highest accuracy and lowest error. This performance surpasses the baselines of direct rotation regression [3, 80], parametric distribution parameters regression [75], and non-parametric distribution estimation [35, 52]. This few-shot training experiment verifies that our SO(3)-equivariant model contributes to superior data efficiency and generalization to unseen rotations.

| Method | Acc@15° | Acc@30° | Rot Err. |
|---|---|---|---|
| Wigner (ours) | **0.6807** | **0.6956** | **22.27°** |
| Euler | 0.0010 | 0.0072 | 132.56° |
| Quaternion | 0.0510 | 0.1629 | 75.95° |
| Axis-Angle | 0.0124 | 0.0815 | 88.66° |
| Rotmat | 0.3909 | 0.5682 | 37.54° |

Table 3: **Comparison of different parametrizations of 3D rotations.** To validate our Wigner-D representation in the frequency domain, we train using various output rotation representations.

| Loss Function | Acc@15° | Acc@30° | Rot Err. |
|---|---|---|---|
| MSE Loss | **0.6807** | **0.6956** | 22.27° |
| L1 loss | 0.6796 | 0.6933 | 22.12° |
| Huber loss | 0.6710 | 0.6873 | **19.26°** |
| Cosine loss | 0.4414 | 0.4978 | 64.29° |
| Geodesic loss | 0.0009 | 0.0071 | 132.65° |

Table 4: **Comparison of different loss functions.** To validate our choice of MSE loss, we experiment with various distance functions between the predicted output and the ground truth.

## 5.5 Ablation Studies & Design Choices

### 5.5.1 SO(3) Parmetrizations

Table 3 shows results validating design choices on ModelNet10-SO(3) with 20-shot training views, using a ResNet-50 backbone. First, we compare the effects of different rotation parametrizations on model performance by changing the prediction head and ground-truth rotations, to verify our proposed Wigner-D compared to other rotation representations. For all cases, we retained the backbone networks and SO(3)-equivariant layers. The only modifications were the output prediction dimension size and the ground-truth rotation representation. Our Wigner-D parametrization outperforms Euler angles (3 dim.), quaternions (4 dim.), axis-angle (4 dim.), and rotation matrices (9 dim.). This demonstrates that frequency domain rotation re-parametrization enables accurate 3D rotations when used with the SO(3)-equivariant spherical CNNs in the frequency domain.

### 5.5.2 Loss Functions

Table 4 compares various loss functions trained on ModelNet10-SO(3) with 20-shot learning using a ResNet-50 backbone. While Huber and L1 losses are alternatives, they do not perform as well as MSE in our context. Cosine loss measures only angle distances between vectors, ignoring magnitude, which is an essential factor in frequency-domain applications. Geodesic loss in the frequency domain is ineffective because it requires separate calculations for each frequency level of the Wigner-D matrix, potentially losing the precision of the original 3D rotation, as we truncate the Fourier basis at a frequency level of 6. Therefore, we choose MSE regression loss for our design choice given its simplicity and effectiveness.

### 5.5.3 Ablation Studies

In Table 5, we experiment with replacing SO(3)-equivariant layers with conventional convolutional layers. Specifically, we use two-layer 1x1 convolutional layers with ReLU activation and a final linear layer with 455 output channels. The results indicate that CNNs without equivariant layers perform poorly, especially in terms of median error, suggesting that using the equivariant networks generalize better to unseen samples. Additionally, the Wigner-D prediction should be paired with an SO(3)-equivariant network to enable reliable 3D rotation prediction in the frequency domain.

|  | Acc@15° | Acc@30° | Rot Err. |
|---|---|---|---|
| ours | **0.6807** | **0.6956** | **22.27°** |
| w.o equivConv | 0.1056 | 0.1308 | 149.25° |
| Random SO(3) | 0.6797 | 0.6946 | 22.16° |
| SuperFibonacci | 0.6785 | 0.6932 | 22.15° |

Table 5: **Comparison of results without the SO(3)-equivariant module and with different SO(3) grids at inference.** The first group shows the results using conventional convolution instead of equivariant convolution. The second group presents the results with different SO(3) grids at inference time. 'ours' denotes the proposed model architecture.

Lastly, we evaluate the impact of different SO(3) grids by switching from a HEALPix grid to a random SO(3) grid and super-Fibonacci spirals [1], which use the same number of SO(3) rotations. Our Wigner-D harmonics predictor performs consistently, regardless of the SO(3) grid sampling type at inference time.

| | SYMSOL I | | | | | | SYMSOL II | | | |
|---|---|---|---|---|---|---|---|---|---|---|
| | avg | cone | cyl | tet | cube | icosa | avg | sphereX | cylO | tetX |
| $\mathcal{L}_{\text{wigner}}$ | 2.54 | 2.42 | 2.68 | 2.93 | 2.67 | 1.99 | -8.88 | 4.51 | -7.64 | -23.52 |
| $\mathcal{L}_{\text{dist}}$ [35] | 3.41 | 3.75 | 3.10 | 4.78 | 3.27 | 2.15 | 4.84 | 3.74 | 5.18 | 5.61 |
| $\mathcal{L}_{\text{wigner}}+\mathcal{L}_{\text{dist}}$ | **4.11** | **4.43** | **3.76** | **5.59** | **3.93** | **2.85** | **6.20** | **6.66** | **5.85** | **6.11** |

Table 6: **Results on SYMSOL I and II [52].** We report the average log likelihood on both parts of the SYMSOL datasets. $\mathcal{L}_{\text{wigner}}$ denotes the results obtained with our Wigner-D regression loss. $\mathcal{L}_{\text{dist}}$ denotes the results using the distribution loss from I-PDF [52], which are the same as the results of I2S [35]. The third row presents the results of joint training using both our regression loss and the distribution loss.

## 5.6 Results on SYMSOL

Table 6 shows symmetric object modeling on the SYMSOL datasets [52]. Compared to the first row and second row [35], our model with only the Wigner-D regression loss derives on sharp modalities, which can be less effective than [35] for symmetric objects in SYMSOL I.

For clearly defined pose cases (e.g., SphereX in SYMSOL II), our Wigner-D loss alone performs well. However, in other SYMSOL II scenarios, the sharp distributions produced by our model can lead to low average log likelihood scores. This metric is particularly harsh on models with sharp peaks, making them vulnerable to very low scores in some failure cases.

In the third row, joint training of our method with the distribution loss [35, 52] achieves better performance than the baseline [35], demonstrating its ability to model symmetric objects. These results highlight the potential of our method in handling complex symmetries and predicting multiple hypotheses. Figure 6 shows the visualization of pose distribution on the SYMSOL I and II datasets.

Most real-world objects have unique, unambiguous poses, validating our single pose regression method (e.g., ModelNet10-SO(3), PASCAL3D+). If the task needs to cover symmetric cases, our model can be modeled with distribution loss [35, 52].

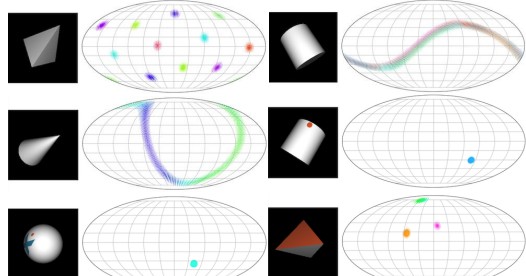

Figure 6: **Visualization of pose distribution on SYMSOL.** The results are obtained by joint training with both our regression loss and the cross-entropy distribution loss [35, 52].

## 6 Conclusion

In this paper, we proposed a novel method for 3D rotation estimation by predicting Wigner-D coefficients directly in the frequency domain using SO(3)-equivariant networks. Our approach effectively overcomes the limitations of existing spatial domain parameterizations of 3D rotations, such as discontinuities and singularities, by aligning the rotation representation with the operations of spherical CNNs. By leveraging frequency-domain regression, our method ensures continuous and precise pose predictions and demonstrates state-of-the-art performance across benchmarks like ModelNet10-SO(3) and PASCAL3D+. Additionally, it offers enhanced data efficiency and generalization to unseen rotations, validating the robustness of SO(3)-equivariant architectures. Our method also supports the modeling of 3D symmetric objects by capturing rotational ambiguities, with further accuracy improvements achievable through joint training with distribution loss. Future work can build on this foundation to explore frequency-domain representations in 3D vision tasks, develop more effective rotation representations for 3D space, and further optimize computational efficiency.

## Acknowledgement

This work was supported by IITP grants (RS-2022-II220959: Few-Shot Learning of Causal Inference in Vision and Language for Decision Making (50%), RS-2022-II220290: Visual Intelligence for Space-Time Understanding and Generation based on Multi-layered Visual Common Sense (45%), RS-2019-II191906: AI Graduate School Program at POSTECH (5%)) funded by Ministry of Science and ICT, Korea.

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

# Appendix / Supplementary Materials

## A  Rotation Representations

### A.1  Rotation Representation in Spatial Domain

We recommend checking out a detailed overview of learning 3D rotations in [54, 58]. Rotations in both 2D and 3D spaces can be represented using various mathematical frameworks, each with its own advantages and limitations, crucial for applications in fields such as computer graphics, robotics, and deep learning. In 2D space, rotation angles can be expressed as following SO(2) representations: angle ($\alpha$) or Trigonometric functions ($\cos(\alpha), \sin(\alpha)$). We recommend a few studies dealing with rotation-equivariance in 2D: group-equivariant networks [10, 65], and its application in image matching [38–40].

In 3D space, Euler angles in $\mathbb{R}^3$ are a representative form of SO(3) representation. Euler angles have 3 DoF consisting of three angles $\alpha, \beta, \gamma \in [-\pi, \pi)$ to describe a 3D rotation (roll, pitch, yaw). 3D rotation matrix can be composed of a fixed sequence rotation using the angles $R(\alpha, \beta, \gamma) = R_z(\alpha)R_y(\beta)R_z(\gamma)$. The standard Euler angles can be divided into 2 forms: Tait-Bryan angles (a.k.a. Cardan angles) which consists of permutations of three items (XYZ, XZY, YXZ, YZX, ZXY, ZYX), and proper Euler angles which starts and ends with rotations around the same axis (ZYZ, ZXZ, XYX, XZX, YZY, YXY), total 12 possible unique sequences. Because of non-uniqueness of Euler angles, existing studies [3, 54, 80] do not encourage the Euler angles as an output representation for 3D rotation prediction in deep neural network.

Quaternions in $\mathcal{S}^3$ is for 4-dimensional complex number to represent a 3D rotation. A quaternion $q$ is composed of one real part and three imaginary parts: $q = w + xi + yj + zk$, where $w, x, y, z$ are real numbers and $i, j, k$ are the fundamental quaternion units. Quaternions can prevent gimbal lock, a problem that occurs with Euler angles where one degree of freedom is lost during 3D rotation.

Axis-angle in $\mathbb{R}^4$ consists of an angle $\theta$ and an axis vector $\mathbf{u}$ in 3D space. To rotate a point $\mathbf{p} \in \mathbb{R}^3$ about the axis $\mathbf{u}$ by an angle $\theta$, you can use Rodrigues' rotation formula: $\mathbf{p}' = \mathbf{p}\cos(\theta) + \mathbf{u} \times \mathbf{p}\sin(\theta) + \mathbf{u}(\mathbf{u} \cdot \mathbf{p})(1 - \cos(\theta))$. The axis-angle representation can be converted into a rotation matrix or a quaternion. Even it has advantages of intuitive form and compact to describe 3D rotation with four parameters, the axis-angle can suffer from singularities in a scenario of angle multiples of $\theta = 2\pi$.

6D representation [3, 80] is a relatively newer concept compared to the traditional representation like Euler angles, rotation matrices and quaternions. A rotation is described by two 3D vectors that are orthogonal to each other, and the rotation matrix can be obtained by Gram-Schmidt orthonormalization (GSO). This representation can be directly predicted and optimized by deep networks because it avoids the non-linear constraints found in quaternions and rotation matrices. Quaternions require maintaining a unit norm, which introduces complexity in ensuring the quaternion remains normalized throughout the optimization process. Therefore, [7, 8, 80] adopt this 6D representation with GSO.

9D representation [3, 41] is direct parametrization of $3 \times 3$ matrix, which can be projected to SO(3) using singular value decomposition (SVD). Predicting 9D representation for rotation matrices involves orthogonality constraints, meaning the rows and columns must remain orthonormal, and determinant constraints, where the determinant must equal +1. These constraints complicate the learning and optimization process, making this representation more suitable for direct prediction and optimization by deep networks. Therefore, 9D representation is less common as a direct method to predict rotation, but [7, 8, 41] use this to mitigate issues associated with discontinuous parameterizations of pose.

### A.2  Rotation Representation in Frequency Domain

3D rotation in the frequency domain is accomplished by manipulating spherical harmonics coefficients. Spherical harmonics $Y_m^l(x) = Y_m^l(\theta, \phi)$ are a function defined on the surface of a sphere, where $\theta$ and $\phi$ are the polar and azimuthal angles, respectively. Here, $l$ represents the degree of the spherical harmonics, $m$ is the order. To ensure computational feasibility, we truncate the degree of harmonics to a finite $L = l_{\max}$. Rotations of spherical functions can be represented by matrices that operate on the coefficients of their harmonics expansion. The rotation of spherical harmonics is expressed via

the shift theorem [46] of spherical harmonics:

$$\Lambda_g Y_m^l(x) = \sum_{|n| \le l} D_{mn}^l(g) Y_n^l(x),  \tag{5}$$

where $\Lambda_g Y_m^l(x)$ denotes the spherical harmonic $Y_m^l(x)$ after rotation by $g$, and $\Lambda_g$ is the rotation operator. Spherical harmonics $Y_m^l(x)$, defined by degree $l$ and order $m$ ($l \ge 0$, $|m| \le l$), use $x$ to represent spherical coordinates. The matrix $U_{mn}^l(g)$ forms part of the irreducible unitary representation of $SO(3)$, showing how each harmonic is transformed under rotation. The sum over all orders $n$ from $-l$ to $l$, $\sum_{|n| \le l}$, shows that $Y_m^l$ is a linear combination of all harmonics of degree $l$.

This rotation of spherical harmonics can be described by the Wigner-D matrix $D_{mn}^l(R)$, which is a unitary matrix that describes the effect of a rotation $R$ on the spherical harmonics basis functions. We can rewrite $U_{mn}^l(g) = D_{mn}^l(\alpha, \beta, \gamma)$. Each element of the matrix represents the amplitude and phase shift that a spherical harmonic $Y_m^l$ undergoes due to the rotation $R$. The rotation $R$ can be specified by Euler angles $\alpha$, $\beta$, and $\gamma$ in the ZYZ-axes configuration. The matrix elements $D_{mn}^l(\alpha, \beta, \gamma)$ can be explicitly expressed as:

$$D_{mn}^l(\alpha, \beta, \gamma) = e^{-im\alpha} d_{mn}^l(\beta) e^{-in\gamma},  \tag{6}$$

where $d_{mn}^l(\beta)$ are the elements of the Wigner (small) d-matrix, which depend only on the angle $\beta$ and are real-valued. These elements capture the intermediate rotation about the Y-axis, where the angle $\beta$ represents the tilt of the axis of rotation. To rotate a spherical harmonic expansion of a function $f$, represented as:

$$f(\theta, \phi) = \sum_{l=0}^{\infty} \sum_{m=-l}^{l} f_m^l Y_m^l(\theta, \phi),  \tag{7}$$

we need to account for the coefficients $f_m^l$ of the expansion. The rotated coefficients $f_m'^l$ are computed:

$$f_m'^l = \sum_{n=-l}^{l} D_{mn}^l(R) f_n^l,  \tag{8}$$

where $D_{mn}^l(R)$ encodes the effect of the rotation on the original coefficients. This transformation preserves the orthonormality and completeness of the spherical harmonics basis, ensuring that the rotated function $f'(\theta, \phi)$ remains a valid representation of the original function $f(\theta, \phi)$ under the rotation $R$. The expansion coefficients $f_m^l$ can be calculated using the original spatial coordinates $(\theta, \phi)$ of the sphere surface according to the spherical harmonic expansion:

$$f_m^l = \int_0^{2\pi} \int_0^{\pi} f(\theta, \phi) \overline{Y_m^l(\theta, \phi)} \sin\theta \, d\theta \, d\phi,  \tag{9}$$

where $\overline{Y_m^l(\theta, \phi)}$ denotes the complex conjugate of the spherical harmonic function $Y_m^l(\theta, \phi)$.

## B   Additional Results

### B.1   Results of Finer Threshold Accuracy

#### B.1.1   ModelNet10-SO(3)

Table A1 presents a comparison of existing methods with different backbones on the ModelNet10-SO(3) dataset[3], highlighting performance across multiple accuracy thresholds and median error. For the ResNet-50 backbone, our method achieves the highest accuracy at 3° (0.422) and 5° (0.640) with a median error of 15.1°, outperforming the existing methods [13, 28, 35, 52]. In particular, our model demonstrates significantly better performance than the strong baselines [28, 35] that use equivariant networks to estimate non-parametric SO(3) healpix distribution at the finer thresholds. Compared

---

[3]We obtained the ModelNet10-SO(3) dataset from the following link: https://github.com/leoshine/Spherical_Regression/blob/master/dataset/ModelNet10-SO3/Readme.md , and the PASCAL3D+ dataset from the following link https://cvgl.stanford.edu/projects/pascal3d.html.

| Method | Backbone | Acc@3° | Acc@5° | Acc@10° | Acc@15° | Acc@30° | Med. (°) |
|---|---|---|---|---|---|---|---|
| Deng *et al.* [13] | ResNet-34 | 0.138 | 0.301 | 0.502 | 0.562 | 0.694 | 31.6 |
| Murphy *et al.* [52] | ResNet-50 | 0.294 | 0.534 | 0.680 | 0.719 | 0.735 | 21.5 |
| Klee *et al.* [35] | ResNet-50 | 0.310 | 0.561 | 0.705 | 0.728 | 0.736 | 15.7 |
| Howell *et al.* [28] | ResNet-50 | - | - | - | - | - | 17.8 |
| ours | ResNet-50 | **0.422** | **0.640** | **0.744** | **0.759** | **0.767** | **15.1** |
| Liao *et al.* [42] | ResNet-101 | - | - | - | 0.496 | 0.658 | 28.7 |
| Mohlin *et al.* [51] | ResNet-101 | 0.164 | 0.389 | 0.615 | 0.693 | 0.757 | 17.1 |
| Yin *et al.* [75] | ResNet-101 | 0.447 | 0.611 | 0.715 | 0.742 | 0.772 | 12.7 |
| Liu *et al.* [44] | ResNet-101 | 0.511 | 0.637 | 0.719 | 0.744 | 0.768 | 12.2 |
| ours | ResNet-101 | **0.513** | **0.688** | **0.763** | **0.773** | **0.780** | **11.9** |

Table A1: **Comparison with finer thresholds on ModelNet10-SO(3).** We compare additional thresholds, including Acc@3°, Acc@5°, and Acc@10°. The tables are organized by the size of the backbone. The scores are averaged across all ten object categories.

| Method | Acc@3° | Acc@5° | Acc@10° | Acc@15° | Acc@30° | Med. (°) |
|---|---|---|---|---|---|---|
| Tulsiani & Malik [61] | - | - | - | - | 0.808 | 13.6 |
| Prokudin *et al.* [57] | - | - | - | - | 0.838 | 12.2 |
| Mahendran *et al.* [45] | - | - | - | - | 0.859 | 10.1 |
| Liao *et al.* [42] | - | - | - | - | 0.819 | 13.0 |
| Mohlin *et al.* [51] | 0.089 | 0.215 | 0.484 | 0.650 | 0.827 | 11.5 |
| Murphy *et al.* [52] | 0.102 | 0.242 | 0.524 | 0.672 | 0.838 | 10.2 |
| Yin *et al.* [75] | 0.134 | 0.292 | 0.574 | 0.714 | 0.874 | 9.3 |
| Klee *et al.* [35] | 0.134 | 0.270 | 0.580 | 0.716 | 0.867 | 9.6 |
| Liu *et al.* [44] | 0.117 | 0.264 | 0.552 | 0.706 | 0.863 | 10.0 |
| Howell *et al.* [28] | - | - | - | - | - | 9.2 |
| ours | **0.153** | **0.310** | **0.595** | **0.754** | **0.892** | **8.6** |

Table A2: **Comparison with finer thresholds on PASCAL3D+.** We compare additional metrics, including Acc@3°, Acc@5°, and Acc@10°, adding on the Table 2. Most baselines [35, 42, 44, 45, 51, 52, 75], including ours, use ResNet-101 backbone networks in this experiment.

to [35], our model achieves 11.2%p, 7.9%p, and 3.9%p higher at the 3°, 5°, and 10° thresholds, respectively. For the ResNet-101 backbone, our method also demonstrates superior performance with the highest accuracy at 3° (0.513) and 5° (0.688), and the lowest median error of 11.9°, compared to [42, 44, 51, 75]. The table includes accuracies at 10°, 15°, and 30° thresholds, where our method consistently shows top performance across these metrics. The scores are averaged across all ten object categories.

### B.1.2 PASCAL3D+

Table A2 presents a comparison of various methods on the PASCAL3D+ dataset, focusing on finer accuracy thresholds (Acc@3°, Acc@5°, and Acc@10°) and median error. The table includes methods that primarily use the ResNet-101 backbone, comparing our approach against several existing methods. Our method achieves the highest accuracies with 0.153 at 3°, 0.310 at 5°, and 0.595 at 10°, with the lowest median error of 8.6°. Compared to the previous state-of-the-art Yin *et al.* [75], our method shows performance improvements of 1.9%p at 3°, 1.8%p at 5°, and 2.1%p at 10°, while also reducing the median error by 0.7°.

### B.1.3 ModelNet10-SO(3) Few-shot Views

Figure A1 shows the results with finer thresholds, Acc@3°, Acc@5°, and Acc@10°, on the ModelNet10-SO(3) few-shot training views, which are additional results to Figure 5. The graphs illustrate that our method outperforms all other methods across all metrics (Acc@3°, Acc@5°, and Acc@10°) and requires fewer training views to achieve high accuracy, even at finer thresholds. This shows that our model is capable of more precise pose estimation with less number of training data, proving the data efficiency of our SO(3)-equivariant harmonics pose estimator. Baseline results [35, 75] were obtained using the source code provided by the authors.

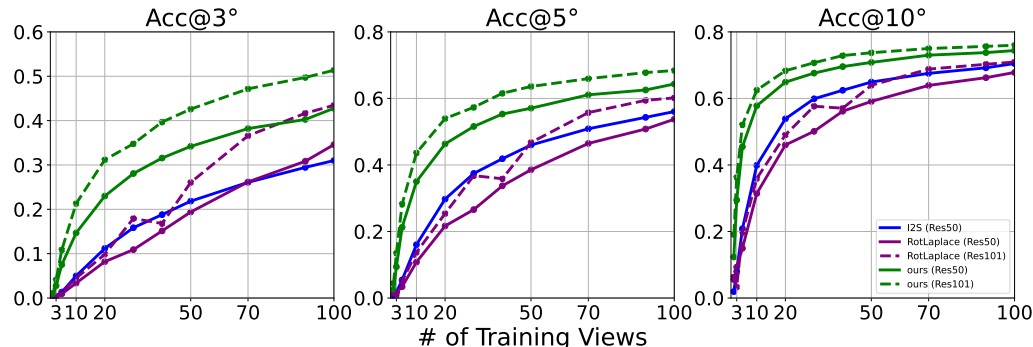

Figure A1: **Results with finer thresholds on ModelNet10-SO(3) few-shot training views.** Results with solid lines denote a ResNet-50 backbone, while dotted lines indicate a ResNet-101 backbone. Our method outperforms all metrics and reduces training views even at finer thresholds. For comparison, the I2S (ResNet-50) model [35] is shown with a blue line, and the RotLaplace (ResNet-50 and ResNet-101) models [75] are depicted with purple solid and dashed lines, respectively.

## B.2 Per-Category Results

### B.2.1 ModelNet-SO(3) Categorical Results

Table A3 provides a comprehensive comparison across 10 object categories on the ModelNet10-SO(3) dataset, including Bathtub, Bed, Chair, Desk, Dresser, TV Monitor, Night Stand, Sofa, Table, and Toilet. Each image is labeled with a single 3D rotation matrix, even though some categories, such as desks and bathtubs, may have ambiguous poses due to symmetry. This poses a challenge for methods that cannot handle uncertainty over orientation. However, in terms of accuracy at 15°, our model consistently achieves the best performance in the desk and bathtub categories, demonstrating robustness against pose ambiguity and symmetry. The table is divided into sections for ResNet-50 and ResNet-101, indicating different network architectures used. For ResNet-50, our method achieves the lowest average median error of 15.1°, with particularly strong performance in 9 categories: bed (2.7°), chair (3.8°), desk (4.2°), dresser (2.7°), tv monitor (2.7°), night stand (3.4°), sofa (7.2°), and toilet (3.0°). For ResNet-101, our model demonstrates the lowest average median error of 11.9°. Although Liu et al. (Uni.) [44] generally obtain better results in terms of median error with the ResNet-101 backbone, our model outperforms Liu et al. (Uni.) [44] on Acc@15° in most cases. This indicates that our model estimates poses correctly at a finer level of detail.

In terms of accuracy at 15°, our method achieves the highest average accuracy of 0.759 for ResNet-50, with best performance in all categories. For ResNet-101 at 15°, our model leads with an average accuracy of 0.773, achieving state-of-the-art performance in 8 out of 10 categories. In terms of accuracy at 30°, our method achieves the highest average accuracy of 0.773 for ResNet-50, with the best results in 9 out of 10 categories. For ResNet-101 at 30°, our model maintains the highest average accuracy of 0.905, with strong results in categories such as Chair, Desk, TV Monitor, and Sofa. Overall, our equivariant harmonics pose estimator demonstrates superior performance across both ResNet-50 and ResNet-101 architectures in terms of both median error and accuracy at different angles. This highlights its effectiveness and robustness across various object categories in the ModelNet10-SO(3) dataset, consistently outperforming other recent methods in most categories and metrics.

### B.2.2 PASCAL3D+ Categorical Results

Table A4 presents comprehensive comparisons on the PASCAL3D+ dataset across 12 categories (Aeroplane, Bicycle, Boat, Bottle, Bus, Car, Chair, Dining Table, Motorbike, Sofa, Train, TV Monitor) using median error and Acc@30° metrics. This benchmark is challenging due to significant variations in object appearances, high variability of natural textures, and the presence of novel object instances in the test set. Our method demonstrates superior performance with the lowest average median error of 8.6° and the highest average accuracy of 0.892, excelling in categories such as "aero," "bike," "car," "chair," "table," and "mbike" in median error, and achieving best performance in 8 out of 12 categories

| | | | ModelNet10-SO(3) categories | | | | | | | | | | |
|---|---|---|---|---|---|---|---|---|---|---|---|---|---|
| | | Methods | Avg. | Bath | Bed | Chair | Desk | Dress | Tv | Stand | Sofa | Table | Toilet |
| Median error (°) | ResNet-50 | Prokudin et al. (2018) [57] | 49.3 | 122.8 | 3.6 | 9.6 | 17.2 | 29.9 | 6.7 | 73.0 | 10.4 | 115.5 | 4.1 |
| | | Zhou et al. (2019) [80] | 41.1 | 103.3 | 18.1 | 18.3 | 51.5 | 32.2 | 19.7 | 48.4 | 17.0 | 88.2 | 13.8 |
| | | Deng et al. (2022) [13] | 32.6 | 147.8 | 9.2 | 8.3 | 25.0 | 11.9 | 9.8 | 36.9 | 10.0 | 58.6 | 8.5 |
| | | Brégier (2021) [3] | 39.9 | **98.9** | 17.4 | 18.0 | 50.0 | 31.5 | 18.7 | 46.5 | 17.4 | 86.7 | 14.2 |
| | | Murphy et al. (2021) [52] | 21.5 | 161.0 | 4.4 | 5.5 | 7.1 | 5.5 | 5.7 | 7.5 | 4.1 | 9.0 | 4.8 |
| | | Klee et al. (2023) [35] | 16.3 | 124.7 | 3.1 | 4.4 | 4.7 | 3.4 | 4.4 | 4.1 | 3.0 | 7.7 | 3.6 |
| | | Howell et al. (2023) [28] | 17.8 | 124.7 | 4.6 | 5.5 | 6.9 | 5.2 | 6.1 | 6.5 | 4.5 | 12.1 | 4.9 |
| | | Ours | **15.1** | 117.4 | **2.7** | **3.8** | **4.2** | **2.7** | **3.8** | **3.4** | **2.5** | **7.2** | **3.0** |
| | ResNet-101 | Liao et al. (2019) [42] | 36.5 | 113.3 | 13.3 | 13.7 | 39.2 | 26.9 | 16.4 | 44.2 | 12.0 | 74.8 | 10.9 |
| | | Mohlin et al. (2020) [51] | 17.1 | 89.1 | 4.4 | 5.2 | 13.0 | 6.3 | 5.8 | 13.5 | 4.0 | 25.8 | 4.0 |
| | | Yin et al. (2023) [75] | 12.2 | **85.1** | 2.3 | 3.4 | 5.4 | 2.7 | 3.7 | 4.8 | 2.1 | 9.6 | 2.5 |
| | | Liu et al. (2023) (Uni.) [44] | 14.6 | 124.8 | **1.5** | **2.8** | **2.7** | **1.5** | **2.6** | **2.4** | **1.5** | **3.9** | **2.0** |
| | | Liu et al. (2023) (Fisher.) [44] | 12.2 | 91.6 | 1.8 | 3.0 | 5.5 | 2.0 | 3.2 | 4.3 | 1.6 | 6.7 | 2.1 |
| | | Ours | **11.9** | 91.9 | 2.3 | 3.3 | 3.2 | 2.1 | 3.4 | 2.7 | 2.2 | 5.4 | 2.6 |
| Acc@15° | ResNet-50 | Prokudin et al. (2018) [57] | 0.456 | 0.114 | 0.822 | 0.662 | 0.023 | 0.406 | 0.704 | 0.187 | 0.590 | 0.108 | 0.946 |
| | | Deng et al. (2022) [13] | 0.562 | 0.140 | 0.788 | 0.800 | 0.345 | 0.563 | 0.708 | 0.279 | 0.733 | 0.440 | 0.832 |
| | | Murphy et al. (2021) [52] | 0.719 | 0.392 | 0.877 | 0.874 | 0.615 | 0.687 | 0.799 | 0.567 | 0.914 | 0.523 | 0.945 |
| | | Klee et al.* (2023) [35] | 0.736 | 0.414 | 0.845 | 0.888 | 0.641 | 0.672 | 0.793 | 0.654 | 0.900 | 0.548 | 0.957 |
| | | Ours | **0.759** | **0.425** | **0.891** | **0.918** | **0.704** | **0.743** | **0.833** | **0.646** | **0.935** | **0.525** | **0.971** |
| | ResNet-101 | Mohlin et al. (2020) | 0.693 | 0.322 | 0.882 | 0.881 | 0.536 | 0.682 | 0.790 | 0.516 | 0.919 | 0.446 | 0.957 |
| | | Yin et al. (2023) | 0.741 | 0.390 | 0.902 | 0.909 | 0.644 | 0.722 | 0.815 | 0.590 | 0.934 | 0.521 | 0.977 |
| | | Liu et al. (2023) (Uni.) [44] | 0.760 | 0.402 | 0.896 | 0.927 | 0.704 | 0.753 | **0.843** | 0.602 | 0.939 | **0.561** | 0.975 |
| | | Liu et al. (2023) (Fisher.) [44] | 0.744 | 0.439 | 0.890 | 0.909 | 0.638 | 0.715 | 0.810 | 0.585 | 0.938 | 0.535 | 0.978 |
| | | Ours | **0.773** | **0.453** | **0.903** | **0.932** | **0.726** | **0.763** | 0.842 | **0.654** | **0.940** | 0.540 | **0.980** |
| Acc@30° | ResNet-50 | Prokudin et al. (2018) [57] | 0.528 | 0.175 | 0.847 | 0.777 | 0.061 | 0.500 | 0.788 | 0.306 | 0.673 | 0.183 | 0.972 |
| | | Deng et al. (2022) [13] | 0.694 | 0.325 | 0.880 | 0.908 | 0.556 | 0.649 | 0.807 | 0.466 | 0.902 | 0.485 | 0.958 |
| | | Murphy et al. (2021) [52] | 0.735 | 0.410 | 0.883 | 0.917 | 0.629 | 0.688 | 0.832 | 0.570 | 0.921 | 0.531 | 0.967 |
| | | Klee et al.* (2023) [35] | 0.736 | 0.427 | 0.848 | 0.915 | 0.642 | 0.672 | 0.819 | 0.565 | 0.902 | **0.555** | 0.964 |
| | | Ours | **0.767** | **0.434** | **0.894** | **0.939** | **0.708** | **0.743** | **0.857** | **0.647** | **0.940** | 0.529 | **0.979** |
| | ResNet-101 | Mohlin et al. (2020) [51] | 0.757 | 0.403 | 0.908 | 0.935 | 0.674 | 0.739 | 0.863 | 0.614 | 0.944 | 0.511 | 0.981 |
| | | Yin et al. (2023) [75] | 0.770 | 0.430 | **0.911** | 0.940 | 0.698 | 0.751 | 0.869 | 0.625 | 0.946 | 0.541 | 0.986 |
| | | Liu et al. (2023) (Uni.) [44] | 0.774 | 0.419 | 0.904 | 0.946 | 0.722 | **0.766** | 0.868 | 0.617 | **0.948** | **0.567** | 0.982 |
| | | Liu et al. (2023) (Fisher.) [44] | 0.768 | **0.460** | 0.898 | 0.934 | 0.694 | 0.738 | 0.859 | 0.615 | **0.948** | 0.544 | **0.987** |
| | | Ours | **0.780** | 0.459 | 0.905 | **0.950** | **0.728** | 0.763 | **0.871** | **0.654** | 0.943 | 0.544 | 0.983 |

Table A3: **Evaluation on ModelNet10-SO(3) by method across different categories.** * denotes reproduced results from the source code provided by authors.

in accuracy at 30°. The results indicate that different methods have varying strengths across different categories, with our SO(3)-equivariant pose harmonics estimation method consistently outperforming others in both accuracy and error metrics, demonstrating its robustness and efficacy.

### B.3 Impact of SO(3) Discretization Sizes and Continuity of Rotations

Table A5 reports the effect of varying the grid size ($Q$) on performance for the ModelNet10 benchmark. We observe comparable results in common evaluation metrics, such as Accuracy at 15 degrees (Acc@15) and 30 degrees (Acc@30), even with a lower grid resolution ($Q$ = 4.6K). A higher resolution grid (18.87M) improves performance under stricter evaluation thresholds. With our chosen grid size of $Q$ = 2.36M, the model achieves strong performance sufficiently, particularly for low-threshold metrics like Acc@3. Table A1 provides additional comparisons to baseline methods.

Therefore, we carefully claim that our learning method focuses on continuous rotations. Our model directly learn the Wigner-D coefficients, which are derived from 3D rotations (Euler angles), without any discretization during the training phase. During inference the use of the SO(3) HEALPix grid serves two purposes: 1) To convert SO(3) rotations from the frequency domain to the spatial domain, and 2) To address pose ambiguity by providing multiple solutions. As a result, we obtain a distribution with very sharp modality. By taking the argmax of this distribution, we achieve sufficient precision in 3D orientation estimation, specifically around 1.5°.

Maintaining continuity in rotations allows our method to deliver more accurate and precise pose predictions, giving us a clear advantage over the methods that experience precision loss from discretization during training. As a result, we achieve consistently high accuracy across different levels of discretization.

| | | PASCAL3D+ categories | | | | | | | | | | | | |
|---|---|---|---|---|---|---|---|---|---|---|---|---|---|---|
| | Method | avg. | aero | bike | boat | bottle | bus | car | chair | table | mbike | sofa | train | tv |
| Median error (°) | Zhou et al. (2019) [80] | 19.2 | 24.7 | 18.9 | 54.2 | 11.3 | 8.4 | 9.5 | 19.4 | 14.9 | 22.5 | 17.2 | 11.4 | 17.5 |
| | Brégier (2021) [3] | 20.0 | 27.5 | 22.6 | 49.2 | 11.9 | 8.5 | 9.9 | 16.8 | 27.9 | 21.7 | 12.6 | 10.2 | 20.6 |
| | Liao et al. (2019) [42] | 13.0 | 13.0 | 16.4 | 29.1 | 10.3 | 4.8 | 6.8 | 11.6 | 12.0 | 17.1 | 12.3 | 8.6 | 14.3 |
| | Mohlin et al. (2020) [51] | 11.5 | 10.1 | 15.6 | 24.3 | 7.8 | 3.3 | 5.3 | 13.5 | 12.5 | 12.9 | 13.8 | 7.4 | 11.7 |
| | Prokudin et al. (2018) [57] | 12.2 | 9.7 | 15.5 | 45.6 | **5.4** | 2.9 | 4.5 | 13.1 | 12.6 | 11.8 | 9.1 | **4.3** | 12.0 |
| | Tulsiani & Malik (2015) [61] | 13.6 | 13.8 | 17.7 | 21.3 | 12.9 | 5.8 | 9.1 | 14.8 | 15.2 | 14.7 | 13.7 | 8.7 | 15.4 |
| | Mahendran et al. (2018) [45] | 10.1 | 8.5 | 14.8 | 20.5 | 7.0 | 3.1 | 5.1 | 9.3 | 11.3 | 14.2 | 10.2 | 5.6 | 11.7 |
| | Murphy et al. (2021) [52] | 10.3 | 10.8 | 12.9 | 23.4 | 8.8 | 3.4 | 5.3 | 10.0 | 7.3 | 13.6 | 9.5 | 6.4 | 12.3 |
| | Yin et al. (2023) [75] | 9.4 | 8.6 | 11.7 | 21.8 | 6.9 | **2.8** | 4.8 | 7.9 | 9.1 | 12.2 | **8.1** | 6.9 | 11.6 |
| | Liu et al. (Uni.) (2023) [44] | 10.2 | 8.9 | 15.2 | 24.9 | 6.9 | 2.9 | 4.3 | 8.7 | 10.7 | 12.8 | 9.3 | 6.3 | 11.3 |
| | Liu et al. (Fisher) (2023) [44] | 9.9 | 9.6 | 12.4 | 22.7 | 7.5 | 3.1 | 4.8 | 9.2 | 8.6 | 13.5 | 8.6 | 6.7 | 11.6 |
| | Klee et al. (2023) [35] | 9.8 | 9.2 | 12.7 | 21.7 | 7.4 | 3.3 | 4.9 | 9.5 | 9.3 | 11.5 | 10.5 | 7.2 | 10.6 |
| | Howell et al. (2023) [28] | 9.2 | 9.3 | 12.6 | **17.0** | 8.0 | 3.0 | 4.5 | 9.4 | 6.7 | 11.9 | 12.1 | 6.9 | **9.9** |
| | ours | **8.6** | **8.3** | **12.1** | 17.2 | 7.9 | 2.9 | **4.2** | **8.1** | 5.5 | **10.4** | 9.3 | 7.1 | 10.7 |
| Acc@30° | Liao et al. (2019) [42] | 0.819 | 0.82 | 0.77 | 0.55 | 0.93 | 0.94 | 0.94 | 0.85 | 0.61 | 0.80 | 0.95 | 0.83 | 0.82 |
| | Mohlin et al. (2020) [51] | 0.825 | 0.80 | 0.75 | 0.53 | 0.95 | 0.96 | 0.96 | 0.78 | 0.62 | 0.87 | 0.93 | 0.77 | 0.84 |
| | Prokudin et al. (2018) [57] | 0.838 | 0.89 | 0.83 | 0.46 | **0.96** | 0.93 | 0.90 | 0.80 | 0.76 | 0.90 | 0.90 | 0.82 | **0.91** |
| | Tulsiani & Malik (2015) [61] | 0.808 | 0.81 | 0.77 | 0.59 | **0.96** | 0.98 | 0.89 | 0.80 | 0.62 | 0.88 | 0.92 | 0.80 | 0.90 |
| | Mahendran et al. (2018) [45] | 0.859 | 0.87 | 0.81 | 0.64 | **0.96** | 0.97 | 0.95 | 0.92 | 0.67 | 0.85 | 0.97 | 0.82 | 0.88 |
| | Murphy et al. (2021) [52] | 0.837 | 0.81 | 0.85 | 0.56 | 0.93 | 0.95 | 0.94 | 0.87 | 0.78 | 0.85 | 0.88 | 0.78 | 0.86 |
| | Yin et al. (2023) [75] | 0.876 | 0.90 | **0.90** | 0.60 | **0.96** | 0.98 | 0.96 | 0.91 | 0.76 | 0.88 | **0.97** | 0.80 | 0.88 |
| | Liu et al. (Uni.) (2023) [44] | 0.827 | 0.83 | 0.78 | 0.56 | 0.95 | **0.96** | 0.93 | 0.87 | 0.62 | 0.85 | 0.90 | 0.81 | 0.86 |
| | Liu et al. (Fisher) (2023) [44] | 0.863 | 0.89 | 0.89 | 0.55 | **0.96** | 0.98 | 0.95 | 0.94 | 0.67 | 0.91 | 0.95 | 0.82 | 0.85 |
| | Klee et al.* (2023) [35] | 0.851 | 0.89 | 0.84 | 0.64 | 0.90 | 0.98 | 0.95 | 0.86 | 0.71 | 0.90 | 0.93 | 0.83 | 0.82 |
| | ours | **0.892** | **0.92** | 0.88 | **0.65** | 0.92 | **1.00** | **0.99** | **0.94** | 0.86 | **0.93** | 0.93 | **0.84** | 0.85 |

Table A4: **Evaluation on PASCAL3D+ by method across different categories.** * denotes reproduced results from the source code provided by authors.

| # of points | width of a bin | Acc3° | Acc5° | Acc10° | Acc15° | Acc30° | Rot Err. |
|---|---|---|---|---|---|---|---|
| 72 | 60° | 0.000 | 0.002 | 0.016 | 0.055 | 0.415 | 45.9° |
| 576 | 30° | 0.002 | 0.014 | 0.122 | 0.396 | 0.765 | 27.1° |
| 4.6K | 15° | 0.026 | 0.116 | 0.615 | 0.750 | 0.767 | 19.7° |
| 36.9K | 7.5° | 0.150 | 0.464 | 0.734 | 0.757 | 0.766 | 17.1° |
| 294.9K | 3.75° | 0.343 | 0.611 | 0.742 | 0.758 | 0.766 | 15.8° |
| 2.36M | 1.875° | 0.424 | 0.641 | **0.746** | **0.760** | 0.767 | **14.7°** |
| 18.87M | 0.938° | **0.443** | **0.646** | **0.746** | 0.759 | **0.768** | 15.0° |

Table A5: **Evaluation by changing the size of the SO(3) grid at inference.** To analyze the sensitivity of discretization on precision ($Q$ of Fig. 4), we vary the recursion levels of the SO(3) HEALPix from 0 to 6. We use a ResNet-50 backbone on ModelNet10-SO(3).

| inference | Acc@15° | Acc@30° | Rot Err. |
|---|---|---|---|
| w/ argmax | 0.7576 | 0.7651 | 12.79° |
| w/ grad ascent | **0.7591** | **0.7660** | **12.43°** |

Table A6: **Comparison of inference methods on pose distribution.** We compare argmax and gradient ascent in the predicted distribution.

| Backbone | Acc@15° | Acc@30° | Rot Err. |
|---|---|---|---|
| ResNet-50 | **0.6807** | **0.6956** | **22.27°** |
| ViT | 0.6384 | 0.6525 | 40.66° |

Table A7: **Results of using transformer instead of convolution.** We train our models by replacing the backbone with Vision Transformer (ViT).

## B.4  Discretised distribution on SO(3)

Table A6 shows the evaluation results using gradient ascent on the predicted SO(3) pose distribution in ModelNet10-SO(3), to fully exploit the distribution prediction in Sec. 4.3 during inference time. While gradient ascent does provide some performance improvement, the increase in inference time outweighs these gains, so argmax is our preferred method for simplicity and fast evaluation.

## B.5  Transformer instead of convolution

Table A7 presents the results when transformers are used as the backbone network instead of the convolutional feature extractor, trained on ModelNet10-SO(3) with 20-shot learning. We trained the

model using a Vision Transformer (ViT) backbone pre-trained by the geometric task of cross-view masked image modeling [66]. Although the ViT is heavier and requires longer training time (1.4x), its performance actually declines. This suggests that convolutional image feature extractor may still be more effective for this 3D orientation estimation task.

## B.6   Searching Frequency Level $L$

Table A8 presents the impact of varying the maximum frequency level $L$ by truncation for efficient SO(3) group convolutions on pose prediction accuracy and median error. The results show that as $L$ increases from 1 to 5, there is a consistent improvement in accuracy metrics. The optimal performance is observed at $L = 5$.

Beyond this point, additional frequency levels do not contribute to improved accuracy and can even degrade performance. When $L > 5$, the accuracy does not improve significantly and starts to fluctuate, with rotation error remaining relatively low up to $L = 10$. However, at $L = 11$, accuracy starts to decline more noticeably. For $L \geq 12$, there is a sharp decline in performance, with Acc@15° dropping to 0.5815 and continuing to decrease, Acc@30° following a similar trend, and rotation error increasing significantly to over 55°. We infer that high frequencies do not improve performance despite the increase in learnable parameters because they lead to overfitting to high-frequency noise. This overfitting occurs when the high-frequency model captures irrelevant noise and patterns in the training data, reducing its generalizability to new, unseen data.

| $L$ | Acc@15° | Acc@30° | Rot Err. |
|---|---|---|---|
| 1 | 0.3637 | 0.5598 | 38.76° |
| 2 | 0.5850 | 0.6839 | 29.00° |
| 3 | 0.6302 | 0.6972 | 26.34° |
| 4 | 0.6670 | 0.6998 | 23.63° |
| 5 | **0.6816** | **0.7014** | 22.23° |
| 6 | 0.6807 | 0.6956 | 22.27° |
| 7 | 0.6731 | 0.6870 | 21.69° |
| 8 | 0.6724 | 0.6848 | **21.46°** |
| 9 | 0.6761 | 0.6884 | 25.34° |
| 10 | 0.6701 | 0.6817 | 21.89° |
| 11 | 0.6625 | 0.6736 | 21.51° |
| 12 | 0.5815 | 0.5956 | 55.23° |
| 13 | 0.5390 | 0.5586 | 55.53° |
| 14 | 0.5228 | 0.5427 | 58.38° |

Table A8: **Results of various number of maximum frequency** $L$ in ModelNet10-SO(3) 20-shot training views.

In conclusion, including higher frequencies ($L > 6$) appears to introduce more noise or overfitting, leading to decreased accuracy and increased rotation error. However, we choose a maximum frequency level $L = 6$ for a fair comparison to [35], and to balance efficiency and accuracy.

## B.7   Justification of the Spherical Mapper

The spherical mapper in Sec. 4.1 maintains the geometric structure of the image when projecting onto the $S^2$ sphere, as detailed in [35]. This method involves lifting the 2D image onto the sphere and converting spherical points using spherical harmonics. Table A9 shows that the spherical mapper outperforms simple Fourier transforms on 2D feature maps.

Using depth information from methods like DepthAnythingv2 for 3D lifting is a good idea and can enhance geometric accuracy. Additionally, centroid ray regression has been explored in research such as [76]. However, incorporating external depth modules increases computational costs and broadens our research scope, so we consider this for future work.

| ModelNet10-SO(3) | | | |
|---|---|---|---|
| projection mode | Acc@15° | Acc@30° | Rot Err. |
| spherical mapper | **0.7590** | **0.7800** | 15.11° |
| MLP mapper | 0.7396 | 0.7457 | **14.98°** |
| PASCAL3D+ | | | |
| Spherical mapper | **0.7535** | **0.8918** | **8.64°** |
| MLP mapper | 0.7283 | 0.8745 | 9.11° |
| ModelNet10-SO(3) 20-Shots | | | |
| Spherical mapper | **0.6807** | **0.6956** | **22.27°** |
| MLP mapper | 0.6446 | 0.6567 | 44.52° |

Table A9: **Validating the design choice of the spherical mapper.** The 'MLP mapper' denotes the Fourier projection, which directly maps image features to harmonics using an MLP, and the 'spherical mapper' denotes our choice of orthographic projection [35].

## B.8   OOD Evaluation

Evaluating out-of-distribution (OOD) performance is generally not the primary focus in the context of this 3D orientation estimation task. However, we have conducted OOD generalization experiments using our proposed method by training the model on different datasets, between ModelNet10 and

| Training Dataset | Evaluation Dataset | Acc@15 | Acc@30 | Rot. Err. |
|---|---|---|---|---|
| ModelNet-SO(3) | ModelNet-SO(3) | 0.7590 | 0.7668 | 15.08° |
| ModelNet-SO(3) | PASCAL3D+ | 0.0004 | 0.0019 | 112.98° |
| PASCAL3D+ | ModelNet-SO(3) | 0.0015 | 0.0086 | 130.44° |
| PASCAL3D+ | PASCAL3D+ | 0.7495 | 0.8965 | 8.92° |

Table A10: **Cross-dataset evaluation** for validating out-of-distribution generalization on ModelNet10-SO(3) and PASCAL3D+ datasets.

| | Klee *et al.* [35] | Yin *et al.* [75] | Liu *et al.* [44] | ours |
|---|---|---|---|---|
| Time (sec. / 1 frame) | 0.0286 | 0.0171 | 3.9960 | **0.0109** |
| GPU memory (GB) | 1.156 | **0.912** | 1.130 | 5.172 |

Table A11: **Comparison of computational cost.** We compare the inference time of one image and GPU memory consumption on ModelNet10-SO(3) test split. To measure the inference time, we average the results of total 18,160 samples of ModelNet10-SO(3) test split.

PASCAL3D+. The results are presented in Table A10. As the results indicate, the model does not perform well when evaluated on an out-of-distribution dataset. Nevertheless, we recognize this as an important area for future research.

### B.9 Computational Cost Analysis

Table A11 presents a detailed comparison of computational cost, focusing on both inference time and GPU memory consumption. The comparison includes the recent baselines; Image2Sphere (2023) [35], RotationLaplace (2023) [75], and RotationNormFlow (2023) [44]. The evaluation is based on the ModelNet10-SO(3) test split, averaging the results from a total of 18,160 samples. We use a machine equipped with an Intel i7-8700 CPU and an NVIDIA GeForce RTX 3090 GPU, utilizing a batch size of 1. The key metrics presented in the table are the inference time per frame (in seconds) and the GPU memory consumption (in gigabytes).

Our model demonstrates the best inference time of 0.0109 seconds per frame, significantly outperforming other models in terms of speed. This efficient inference time translates to approximately 92.5 frames per second (FPS) for an image size of 224x224, making our model suitable for real-time applications. However, this performance comes at the cost of higher GPU memory consumption, which is recorded at 5.172 GB. Since our model performs all operations on the GPU, we achieve a temporal advantage in inference time, despite having many overlapping modules with Klee *et al.* [35], whose model performs some computations on the CPU. In summary, our model achieves the best inference time, facilitating real-time application potential, by trading off increased GPU memory consumption.

### B.10 Experiment of Statistical Significance

Table A12 presents the results of a 5-trial experiment to evaluate the training sensitivity of our models, with ResNet-50 and ResNet-101, on the ModelNet10-SO(3) 20-shot training views. The table lists individual trial results, along with the average ($\mu$) and standard deviation ($\sigma$) for each metric. The standard deviation values ($\sigma$) for both backbones are relatively small across all metrics, suggesting that the models yield consistent results over multiple trials. For instance, the standard deviation of Acc@3° for ResNet-50 is 0.0047, and for ResNet-101, it is 0.0052, which are both quite low. This low variance indicates that the training results are stable and reproducible. These findings highlight the robustness and reliability of the training process and the effectiveness of ResNet-101 for the given task.

## C  Training Details

We utilize the `e3nn` library [22] for $S^2$ and SO(3) convolutions for efficient handling of both Fourier and inverse Fourier transforms, `healpy` [24, 81] for HEALPix grid generation, and `PyTorch` [53]

|  | Trial | Acc@3° | Acc@5° | Acc@10° | Acc@15° | Acc@30° | Med Err. |
|---|---|---|---|---|---|---|---|
| **ResNet-50** | 1 | 0.2299 | 0.4632 | 0.6490 | 0.6807 | 0.6956 | 22.27° |
|  | 2 | 0.2336 | 0.4687 | 0.6503 | 0.6793 | 0.6944 | 21.97° |
|  | 3 | 0.2292 | 0.4691 | 0.6522 | 0.6827 | 0.6963 | 21.00° |
|  | 4 | 0.2252 | 0.4593 | 0.6465 | 0.6802 | 0.6949 | 28.06° |
|  | 5 | 0.2375 | 0.4699 | 0.6535 | 0.6847 | 0.6978 | 18.93° |
|  | $\mu$ | 0.2311 | 0.4660 | 0.6503 | 0.6815 | 0.6958 | 22.49° |
|  | $\sigma$ | 0.0047 | 0.0046 | 0.0027 | 0.0022 | 0.0013 | 3.92 |
| **ResNet-101** | 1 | 0.3108 | 0.5418 | 0.6877 | 0.7099 | 0.7214 | 20.91° |
|  | 2 | 0.3115 | 0.5414 | 0.6848 | 0.7076 | 0.7184 | 20.77° |
|  | 3 | 0.3054 | 0.5413 | 0.6843 | 0.7067 | 0.7188 | 21.61° |
|  | 4 | 0.3158 | 0.5451 | 0.6875 | 0.7101 | 0.7216 | 20.87° |
|  | 5 | 0.3027 | 0.5366 | 0.6842 | 0.7088 | 0.7204 | 16.83° |
|  | $\mu$ | 0.3092 | 0.5412 | 0.6857 | 0.7086 | 0.7201 | 20.20° |
|  | $\sigma$ | 0.0052 | 0.0030 | 0.0018 | 0.0015 | 0.0015 | 1.91 |

Table A12: **Experiment of 5-trials training of our model for statistical significance** on ModelNet10-SO(3) 20-shot training views. $\mu$ denotes the average, and $\sigma$ denotes the standard deviation.

for model implementation. We use a machine with an Intel i7-8700 CPU and an NVIDIA GeForce RTX 3090 GPU. With a batch size of 64, our network is trained for 50 epochs on ModelNet10-SO(3) taking 25 hours, and for 80 epochs on PASCAL3D+ taking 28 hours. We start with an initial learning rate of 0.1, which decays by a factor of 0.1 every 30 epochs. We use the SGD optimizer with Nesterov momentum set at 0.9. Unlike baselines that encode object class information via an embedding layer during training [35] and both training and testing [44, 75], our model does not use class embeddings, maintaining a class-agnostic framework during both training and testing. Additionally, we train a single model for all categories in each dataset, unlike [7], which trains separate models for each class.

## D   Baselines of Single-View Pose Estimation

We compare our method against competitive single-view SO(3) pose estimation baselines including regression methods and distribution learning methods. Zhou *et al.* [80] predict 6D representations using Gram-Schmidt orthonormalization processes for 3D rotations, analyzing the discontinuities in rotation representations. Brégier [3] extends deep 3D rotation regression with a differentiable Procrustes orthonormalization, which maps arbitrary inputs from Euclidean space onto a non-Euclidean manifold. Tulsiani and Malik [61] train a CNN using logistic loss to predict Euler angles. Mahendran *et al.* [45] predict three Euler angles using a classification-regression loss to estimate fine-pose while modeling multi-modal pose distributions. Liao *et al.* [42] also predict Euler angles using a classification-regression loss by introducing a spherical exponential mapping on n-spheres at the regression output.

On the other hand, the other baselines are generating probability distributions for estimating SO(3) pose. Prokudin *et al.* [57] represents rotation uncertainty with a mixture of von Mises distributions over each Euler angle, while Mohlin *et al.* [51] predicts the parameters for a matrix Fisher distribution. Deng *et al.* [13] predict multi-modal Bingham distributions. Murphy *et al.* [52] trains an implicit model to generate a non-parametric distribution over 3D rotations. Yin *et al.* [74, 75] predict the parameter of SO(3) parametric distribution using matrix-Fisher distribution and rotation Laplace distribution, respectively. Klee *et al.* [35] predicts non-parametric distribution with equivariant feature prediction by orthographic projection, and Howell *et al.* [28] extends to construct neural architectures to satisfy SO(3) equivariance using induced and restricted representations. Liu *et al.* [44] use discrete normalizing flows for rotations to learn various kinds of distributions on SO(3). Results are from the original papers when available.

# E  Qualitative Results

Figures A2 and A3 show qualitative results randomly selected from ModelNet10-SO(3) and PAS-CAL3D+, respectively. For visualization, we display distributions over SO(3) as proposed in I-PDF [52]. To illustrate the SO(3) distribution, we use the Hopf fibration to visualize the entire space of 3D rotations [72]. This approach maps each point on a great circle in SO(3) to a point on the discretized 2-sphere and uses a color wheel to indicate the location on the great circle. Essentially, each point on the 2-sphere represents the direction of a canonical z-axis, and the color represents the tilt angle around that axis. To depict probability score, we adjust the size of the points on the plot. Lastly, we present the 2-sphere's surface using the Mollweide projection.

**Comparison of pose visualization.** Figures A4 and A5 show a comparison of pose visualizations on ModelNet10-SO(3) and PASCAL3D+, respectively. This visualization method is the same to those used in Figures A2 and A3. We compare our model to the I2S [35] baseline. The numbers next to "Err" above the input images represent the error in degrees between the model's predicted pose and the ground truth (GT) pose. These results demonstrate that our model provides more accurate and precise pose estimations, even in cases where the I2S baseline fails. Additionally, on the PASCAL3D+ benchmark, which includes objects captured in real-world scenarios, our model consistently shows correct pose estimations, particularly in challenging scenarios where the I2S baseline struggles.

# F  Limitation

Our proposed method significantly advances 3D rotation estimation accuracy; however, a notable challenge in pose estimation is the issue of pose ambiguity, particularly for objects with symmetrical features or those viewed from certain angles, e.g., bathtub category in ModelNet10-SO(3). Despite high accuracy, our method can suffer from significant errors due to the loss of spatial information when projecting 3D data onto spherical harmonics. Future work could integrate additional contextual or spatial information to mitigate these ambiguities, improve reliability, and enhance the model's robustness in diverse scenarios. Additionally, while the mathematical rigor of using spherical harmonics and Wigner-D coefficients supports the model's success and improves interpretability through equivariant networks, further exploration is needed to make the model more interpretable. Finally, the computational cost associated with Wigner-D coefficients and SO(3)-equivariant networks should be improved to enhance practicality for real-time applications and deployment on devices with limited processing power.

# G  Broader Impacts

The method proposed in this paper has several potential positive societal impacts. First, it can enhance robotics and automation. Accurate 3D pose estimation is crucial for these fields, and improved accuracy can lead to more efficient and safer robotic systems in manufacturing, healthcare, and service industries. Second, it can significantly advance augmented reality (AR) applications by providing more precise alignment of virtual objects with the real world, which can be beneficial in education, gaming, and industrial design. Third, the method can improve autonomous vehicles, which rely on precise 3D pose estimation to understand their environment, contributing to safer and more reliable autonomous driving systems. Finally, in medical imaging, accurate pose estimation can improve the analysis and interpretation of complex 3D data, aiding in diagnosis and treatment planning.

However, the paper also suggests potential negative societal impacts. Improved pose estimation techniques could be used in surveillance systems, leading to privacy concerns if deployed without proper regulations and oversight. Enhanced 3D pose estimation could be exploited to create more realistic deepfakes, contributing to the spread of disinformation and manipulation. Deployment of these technologies could inadvertently reinforce existing biases if the training data is not representative of diverse populations, leading to unfair treatment of specific groups in applications like security and hiring. Additionally, the misuse of accurate pose estimation in security-sensitive areas, such as military applications or unauthorized monitoring, could pose significant risks.

To address these potential negative impacts, several mitigation strategies could be implemented. Controlled release of models and methods to ensure ethical and responsible use is one approach.

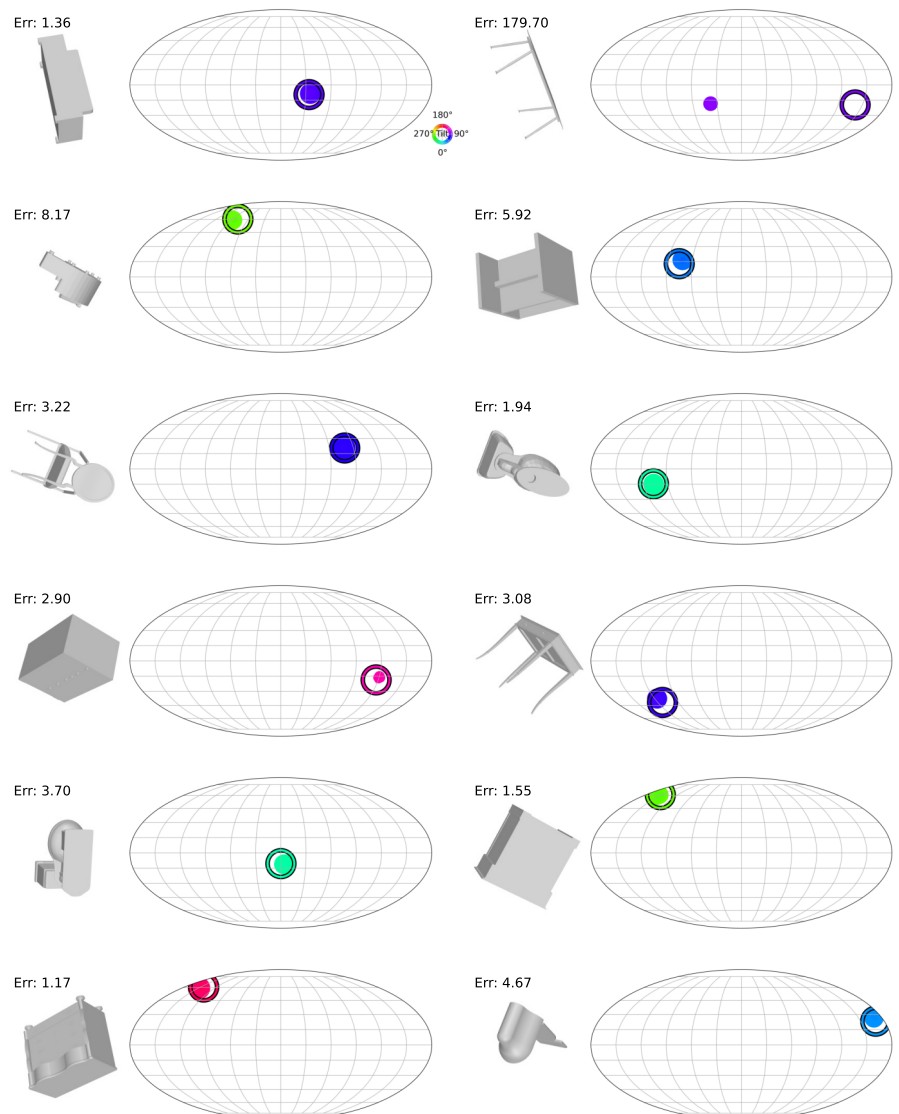

Figure A2: Randomly selected qualitative results of pose estimation on ModelNet10-SO(3) using our SO(3) equivariant harmonics pose estimator. The error value, indicating the difference between the estimated and ground truth orientations in degrees, is labeled above each plot. Most images with clearly posed objects in the input image show an error of 10°or less, demonstrating high accuracy of the pose estimation algorithm. The example in the first row, second column, shows a significant error of 179.70°. This high error is attributed to the ambiguity in pose information, as the projection of the 3D object causes a loss of spatial information, resulting in larger discrepancies between the ground truth and estimated poses. Other examples with low errors, such as the top-left corner (Err: 1.36°) and second row, second column (Err: 1.94°), indicate successful pose estimations.

Regular audits to ensure the training data and algorithms do not propagate biases are also crucial. Implementing robust privacy protection measures can safeguard individual privacy in applications involving surveillance. Developing complementary technologies to detect and mitigate the effects of deepfakes and other forms of disinformation is another necessary step.

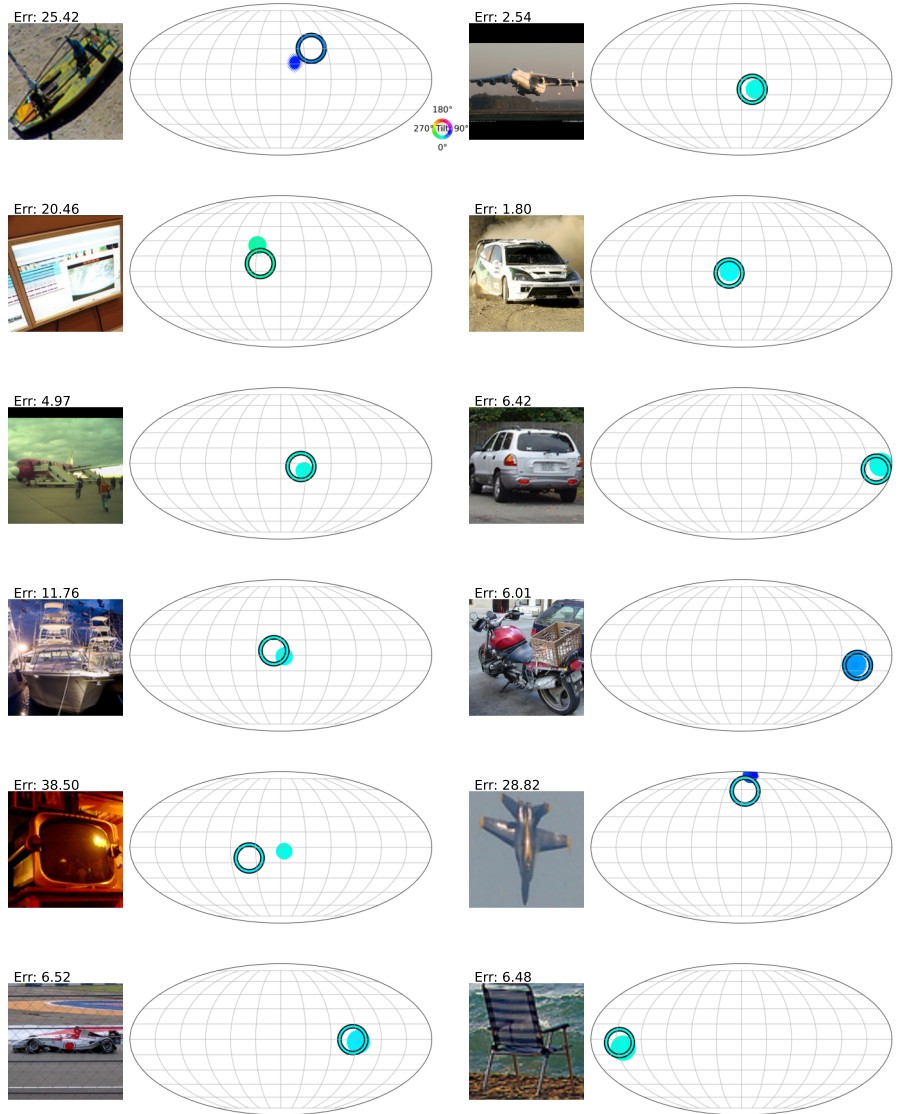

Figure A3: Randomly selected qualitative results on PASCAL3D+ using our SO(3) equivariant pose harmonics estimator. The error value, indicating the difference between the estimated and ground truth orientations in degrees, is labeled above each plot. Most images with clearly posed objects in the input image show an error of 10°or less, demonstrating high accuracy of the pose estimation algorithm. For example, the airplane in the first row, second column, shows a low error of 2.54°, indicating precise pose estimation. However, some objects, like the monitor (Err: 38.50°), airplane (Err: 28.82°) in the fifth row, exhibit larger errors, possibly due to pose ambiguity in the input image by symmetry. The variability in errors across different objects highlights the our model's performance variability depending on the object's shape and the clarity of pose information in the input image.

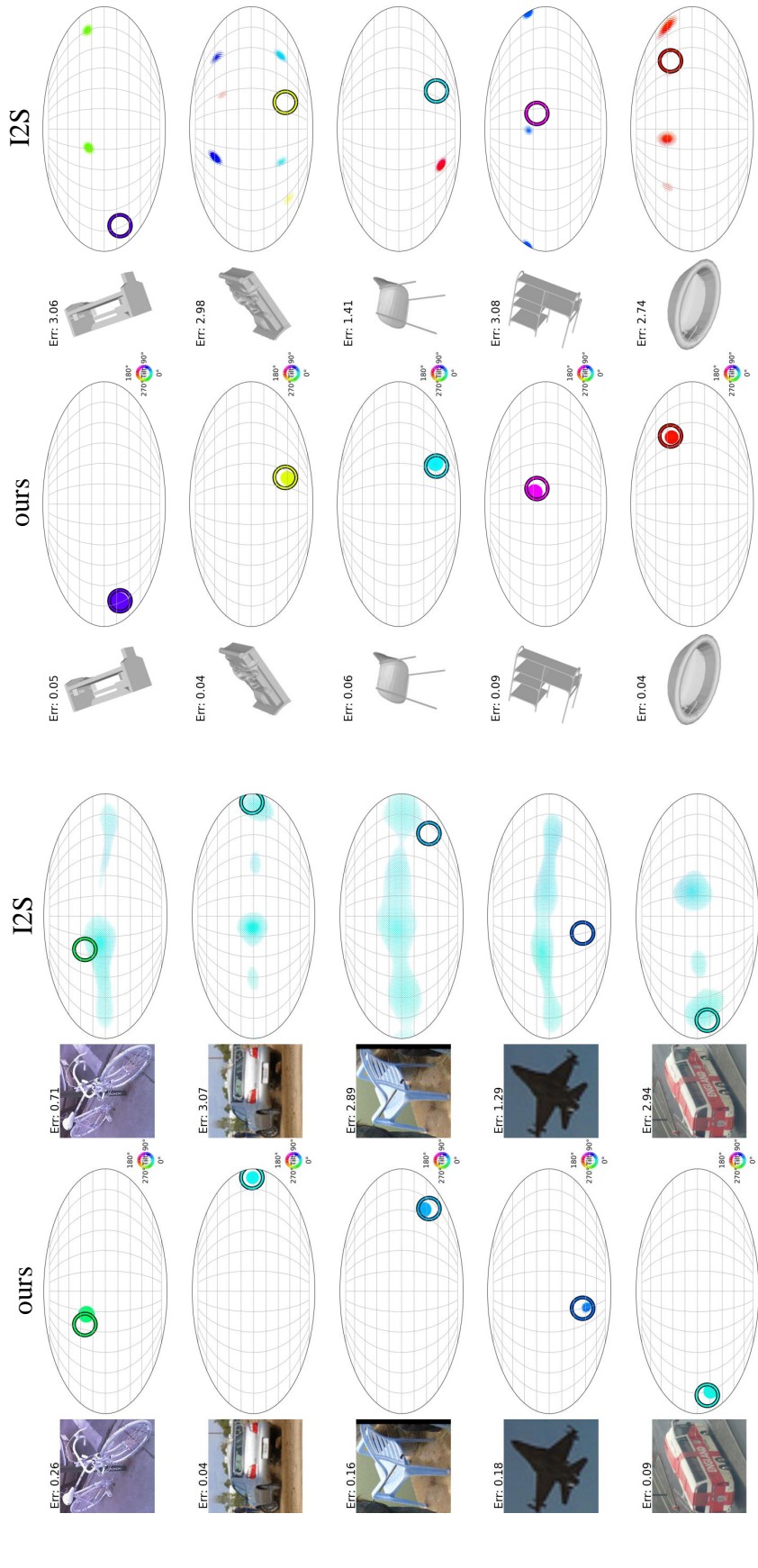

Figure A4: Comparison of pose visualization with I2S [35] baseline on ModelNet10-SO(3).

Figure A5: Comparison of pose visualization with I2S [35] baseline on PASCAL3D+.

