# OpenReview forum: "3D Equivariant Pose Regression via Direct Wigner-D Harmonics Prediction"
_NeurIPS.cc/2024/Conference — NeurIPS 2024 poster_

### Official Review · Reviewer_f5v8 · 2024-07-01

**Soundness:** 3
**Presentation:** 2
**Contribution:** 2
**Rating:** 6
**Confidence:** 5

**Summary:**

This paper proposes a method for object orientation estimation from images.  The method predicts object orientation in the frequency domain of SO(3), by using SO(3) equivariant layers that operate on coefficients of Wigner-D matrices.  The network is trained with a MSE loss between the predicted Wigner-D coefficients and the Wigner-D coefficients associated with a ground truth rotation label.  The network is evaluated on two datasets, PASCAL3D+ and ModelNet10-SO(3), and outperforms all baselines in terms of rotation precision and accuracy.  Additionally, an ablation study is performed which demonstrates that the proposed output formulation is better than traditional rotation formulations, like quaternions, euler angles and rotation matrices, and that the method is robust to the sampling strategy used at inference time.

**Strengths:**

- The paper introduces a new way to train Wigner-D coefficients for accurate object pose prediction.  They apply a weighted MSE loss between the predicted coefficients and  ground truth coefficients.
- The proposed approach achieves SOTA performance on the PASCAL3D+ and ModelNet10-SO(3) datasets.
- The experimental section is thorough.  Competitive baselines are included and statistical data is reported for the proposed method.  The authors include a new variation on the ModelNet10-SO(3) dataset, which evaluates the sample efficiency of the method and baselines.

**Weaknesses:**

- The contribution is interesting, but minimal compared to existing work.  The neural network model and evaluation scheme used in this paper was introduced in [1].  However, this is not entirely clear the way the paper is written.  For instance, Sections 4.2, 4.3, 4.4, and 4.6 are all describing specific techniques from [1].  It would be easier to understand the contributions specific to this paper's network if these sections were moved to Background.
- The second contribution of the paper, as stated, is confusing.  What is meant by the "intrinsic properties of SO(3) equivariant representations"?  Also, the claim "structurally guarantee 3D rotational equivariance" is not correct.  It is not possible to guarantee 3D rotational equivariance of the network when the input is an image.
- The core contribution of the paper is the use of a regression loss applied to learned Wigner-D coefficients.  However, this aspect of the method is not examined closely.  There is no information of how the values of $w_l$ are defined in Eqn. 5, nor any experiment justifying why MSE is preferred.  The contribution would be strengthened by a comparison of other sensible loss functions, an experiment that showed the sensitivity of the method to the values of $w_l$.  Additionally, it is not clear how the proposed loss is affected by object symmetries.  Could the authors explain why regression on Wigner-D coefficients does not suffer from inaccurate predictions when the orientation is ambiguous (e.g. tables have 180 degree symmetry in ModelNet10-SO(3))?
- The ablation experiment in Section 5.5 needs to be described in more detail.  How exactly was the method modified to produce euler, quaternion, etc. outputs?  Were SO(3) equivariant layers still used?
-  Some of the writing could be improved for clarity.  In Section 1, "Accurate determination of 3D orientation presents a more complex problem than translation due to the intricacies of rotation symmetries and the high dimensionality of pose space" is confusing: how is 3D orientation higher dimensional than 3D translation?  Also, "SO(3)-equivariance is crucial for accurate pose estimation ..." is not entirely true, as evidenced by the competitiveness of non-equivariant baselines in Tables 1&2.
- Figure 1, some of the images are taken from other sources without attribution (e.g. the healpix grid is from https://healpix.sourceforge.io



[1] Klee, David M., et al. "Image to sphere: Learning equivariant features for efficient pose prediction." arXiv preprint arXiv:2302.13926 (2023).

**Questions:**

- Did you explore evaluating the model with a higher resolution grid or with gradient ascent?  Given that the regression loss can be precise, the model may be able to achieve even lower rotation error with more grid points.
- The regression loss is very effective at achieving high precision, but seems to lose the ability to model object symmetries or uncertainty (according to qualitative results in Figure A2).  Is there a way to combine the benefits of high precision and distribution modeling?  Other pose works have used a classification-regression framework [1], and it would be interesting to see if this idea could be applied here.  For instance, apply the classification loss over a HEALPix grid first, then apply the proposed regression loss if the prediction is close.
- In Appendix C.6, it seems like the frequency level is set for both the S2 and SO(3) convolutions in the network.  Did you try only modifying the frequency for the output of the network?  The issue with higher frequencies reducing performance may be due to the non-linearity (ReLU in spatial domain), which could be a bit unstable with high frequency signals.  Some other works have used gating functions for non-linearities on Wigner D coefficients [2].
- From the introduction, "in the context of spherical CNNs, ..., the 3D rotation parametrization in the spatial domain is inadequate because these SO(3) equivariant networks operate in the frequency domain."  What do the authors mean by inadequate?  Clearly, conversion from frequency to spatial domain is possible (the proposed method does so in its non-linearity layers).

[1] Liao, Shuai, Efstratios Gavves, and Cees GM Snoek. "Spherical regression: Learning viewpoints, surface normals and 3d rotations on n-spheres." Proceedings of the IEEE/CVF Conference on Computer Vision and Pattern Recognition. 2019.

[2] Liao, Yi-Lun, and Tess Smidt. "Equiformer: Equivariant graph attention transformer for 3d atomistic graphs." arXiv preprint arXiv:2206.11990 (2022).

**Limitations:**

The limitations are discussed in Appendix F.

What is meant by, "our method can suffer from significant errors due to the loss of spatial information when projecting 3D data onto spherical harmonics"?  What 3D data?  Where are these significant errors described in the experimental results or discussion?

The authors mention that, "computational cost associated with Wigner-D coefficients and SO(3) equivariant networks should be improved...";
this is a bit vague and contradicts Table A6, which shows the proposed method has the fastest frame rate during evaluation.

---

> ### Author Rebuttal · Authors · 2024-08-07
>
> We thank reviewer f5v8 for constructive comments and suggestions.
>
> **[W1. Clarifying contributions]**
>
> Please see our general response above
>
> **[W2. Claims of structural guarantee of 3D rotational equivariance]**
>
> The statement "structurally guarantee 3D rotational equivariance" is indeed misleading. While the network can use components like spherical convolutions and Wigner-D matrix predictions to maintain equivariance, perfect 3D rotational equivariance cannot be guaranteed with 2D image inputs, as they inherently lack full 3D structural information. The network approximates 3D equivariance using the spherical mapper to handle 3D rotations effectively, but absolute equivariance is not achievable with 2D data. We will clarify that the network aims to approximate 3D rotational equivariance as closely as possible given these constraints.
>
> **[W3. symmetry, other losses, $w_l$]**
>
> Please see our response above
>
> **[W4. Detailed configurations of Table 3 ablation studies]**
>
> Table 3 in Section 5.5 demonstrates the effectiveness of our proposed Wigner-D matrix compared to other rotation representations in the spatial domain (Euler angles, Quaternion, axis-angle, rotation matrix). For all cases, we retained the backbone networks and SO(3) equivariant layers. The only modifications were the output prediction dimension size and the ground-truth rotation representation.
>
> **[W5-1. Clarification on the complexity of 3D orientation and translation]**
>
> We acknowledge that the original statement might be confusing.  To clarify, we can represent the 3D translation and orientation:
>
> * 3D translation: movement along the x, y, and z axes. (3 dims)
> * 3D orientation: involves rotations around these axes, represented by Euler angles (3 dims), quaternions (4 dims), or rotation matrices (9 dims).
>
> We will update our manuscript to improve clarity: 3D orientation is more complex due to rotational symmetries and the non-linear nature of rotations. Unlike translation, rotations present challenges such as gimbal lock and the need for continuous representation without singularities.
>
>
> **[W5-2. Overstatement of SO(3) equivariance]**
>
> The claim "SO(3)-equivariance is crucial for accurate pose estimation" might be overstated. While it can enhance performance and robustness, it is not the only way to achieve competitive results. As shown by the non-equivariant baselines in Tables 1 and 2, alternative approaches also deliver strong performance. We will revise the manuscript to reflect this nuanced perspective.
>
>
> **[W6. Attribution of images in figure 1]**
>
> We apologize for the oversight in attributing images. We will revise the figure to clearly credit all original sources.
>
> **[Q1. Evaluation of model with higher resolution grid and gradient ascent]**
>
> Table R2 shows the impact of changing the grid resolution $Q$ on performance. The higher resolution (18.87M) of the SO(3) grid improves the performance at a lower threshold at inference time.
>
> Table R5 shows the evaluation results using gradient ascent on the pose distribution.  While gradient ascent does provide some performance improvement, the increase in inference time outweighs these gains, so argmax is our preferred method for simplicity and fast evaluation.
>
> **[Q2. Combining high precision and distribution modeling]**
>
> Our method currently struggles to model object symmetries and uncertainty. However, Table R1 demonstrates that combining our approach with a distribution loss based on cross-entropy [48, 34] is effective for symmetric object modeling, and a classification-regression framework [38] can be an alternative solution for distribution modeling.
>
>
> **[Q3. Adjusting frequency levels and mitigating high-frequency instability]**
>
> Increasing the frequency level at the network's output is challenging due to the need to define the maximum frequency level of spherical harmonics at the model initialization.  Adjusting the final output frequency can help mitigate performance drops at high frequencies caused by non-linearity. Using gating functions on Wigner-D coefficients in Equiformer (Liao and Smidt, ICLR 2023) could address instability issues.
>
>
> **[Q4. Clarification on spatial domain parametrization in spherical CNNs]**
>
>
> The term "inadequate" refers to the practical challenges and inefficiencies of using spatial domain parametrizations directly within spherical CNNs. These challenges arise because spherical CNNs naturally operate in the frequency domain, where rotations are represented and manipulated more efficiently using spherical harmonics and Wigner-D matrices.
>
> Although conversion between the frequency and spatial domains is possible (by inverse Fourier transform), maintaining consistent parametrization in the frequency domain is more straightforward for the SO(3)-equivariance with spherical CNNs and proves to be more effective, as demonstrated in Table 3 of the main paper.
>
>
>
>
> **[L1. Clarifying the loss of spatial Information in spherical harmonics projection]**
>
> In Section F, the statement about significant errors refers to the loss of spatial information due to truncating the frequency level of spherical harmonics for efficiency. "3D data" refers to data points on the 2-sphere. These errors, not explicitly detailed in the experimental results, arise from truncating higher frequency components, leading to a loss of spatial detail. We will update the manuscript to clarify this point and provide a more precise explanation.
>
> **[L2. Clarifying computational costs in the limitation section]**
>
> In Section F, we note that while our method achieves the best inference time (Table A6), there is room for improvement in space complexity. The computational cost, due to the complexity of Wigner-D coefficients and convolutions in SO(3)-equivariant networks, remains high. These operations require handling high-dimensional representations and specialized mathematical processes. Therefore, reducing memory usage and computational load is a target for future enhancement.

---

> ### Comment · Reviewer_f5v8 · 2024-08-08
>
> Thank you for the response.  I appreciate all the new results presented in the PDF, and the decision to revise the background/related work sections.
>
> It is exciting to see the proposed MSE loss can be combined with the Cross-Entropy loss from I2S to achieve both high precision and accurate distributions.
>
> Regarding R3, what exactly is the takeaway from this plot?  That the model puts more weight on higher-frequencies early on in training?  Is this something that could be addressed with a weighted MSE loss to boost performance in the very low data regime?
>
> Regarding R4, it would be nice to understand the pitfalls of the MSE loss over Wigner-D coefficients.  For instance, could the gradient of the MSE loss push the model away from the ground truth rotation in some cases?  This doesn't need to be addressed here but it would be interesting to study in the future.
>
> Regarding R5 and R2, it is interesting that there is a sweet spot in terms of how precise the model can be.  Do you think this is due to the limited precision of the truncated Fourier series?
>
> Given the proposed revisions and the new results, I updated my rating.

---

> > ### Author Response · Authors · 2024-08-11
> > **Response of the Official Comment by Reviewer f5v8**
> >
> > Thank you for updating your rating by reviewing our rebuttal and revision proposal of the background/related work sections, and thank you for your thoughtful feedback.
> >
> > * * *
> > **Takeaway from the plot in Table R3**: The plot demonstrates that the model consistently assigns greater weight to higher frequencies ($y$-axis and value), which correspond to more complex rotational components in the Wigner-D matrix. This suggests that the model effectively captures and prioritizes these complex aspects, regardless of the number of training views ($x$-axis).
> >
> > Additionally, the plot further indicates that the weighted MSE loss plays a key role in enabling the model to focus on more complex rotations early stage in the training process, ensuring that these components are adequately emphasized even when data is limited (very low data regime).
> >
> > * * *
> > **Understanding the pitfalls of the MSE Loss in Table R4**: Yes, exploring whether the MSE loss could potentially push the model away from the ground truth rotation is indeed a valuable direction for future research. However, at this point, we have not observed or verified instances where the model moves away from the ground truth rotation due to the MSE loss.
> >
> > * * *
> > **The “sweet spot” in terms of the model precision in Tables R2 and R5**:
> > The "sweet spot" observed in Tables R2 and R5 is indeed an interesting phenomenon. However, we do not attribute this is due to the limited precision of the truncated Fourier series. Instead, it seems to be more related to the trade-off between computational cost and performance.
> >
> > In Table R2, as the SO(3) grid precision increases, we observe that model accuracy improves and then saturates. This suggests that beyond a certain point, further increasing the precision (e.g. the number of points, $Q$) yields diminishing returns in accuracy relative to the additional computational cost. For example, we identified a good trade-off at $Q$=2.36M points (6th row), where further increasing to $Q$=18.87M points (7th row) improves accuracy at finer error thresholds (Acc@3° ) but also significantly increases space complexity.
> >
> > Similarly, in Table R5, employing gradient ascent does enhance model accuracy and reduces error, particularly for more precise pose estimations. However, this also results in a significant increase in inference time. Thus, the observed "sweet spot" in precision is more likely due to the inherent trade-offs in computational resources and the model's ability to effectively utilize them, rather than limitations imposed by the truncated Fourier series.

---

### Official Review · Reviewer_HCET · 2024-07-06

**Soundness:** 2
**Presentation:** 2
**Contribution:** 1
**Rating:** 3
**Confidence:** 4

**Summary:**

The paper tackles 3D pose estimation from a single image. It builds on Image2Sphere which maps CNN features to the sphere which are then processed by spherical CNNs to produce a distribution over SO(3). The submission proposes using an MSE loss function in the spectral domain instead of cross-entropy in the spatial domain. It outperforms the baselines on pose estimation on ModelNet10-SO(3) and Pascal3D+.

**Strengths:**

S1) There could be value in studying losses and other operations in the spectral domain instead of spatial.

S2) The method seems to outperform the baselines in the attempted benchmarks.

**Weaknesses:**

W1) The paper seems to be a lightly modified version of Image2Sphere [1], but this is not properly acknowledged in the writing. Following Figure 2 and Section 4, the feature extraction, spherical mapper, Fourier transformers and pose predictor seems to be the same, the only difference being that in the submission, the last IFT is skipped and the loss is computed in the spectral domain during training (at test time it still needs the IFT so it is the same as Image2Sphere as far as I understand). Although Image2Sphere [1] is cited, the method section mostly describes the same things which implies they are new contributions. Please rewrite it focusing on the new contributions and being explicitly about what comes from prior work.

W2) The main difference with respect to Image2Sphere is the loss being computed between Wigner-D coefficients instead of spatially. However how this is actually done is poorly explained.

a) How does one L245 "convert the GT 3D rotations from Euler angles (...) to the Wigner-D matrices"? The text seems to imply that the Wigner-Ds are evaluated at the ground truth rotation, but then the loss in Eq (5) does not seem to make sense since the output of the model are the coefficients associated to each D matrix. I think what should be done is inverting a function on SO(3) that is an impulse on the ground truth pose but there is no mention of it in the paper.

b) What is the "similarity between Wigner-D coefficients and an SO(3) grid" is L263? It seems that this somehow outputs a distribution on SO(3) that is queried spatially, so it seems like it should be an inverse SO(3) Fourier transform?

W3) Experiments on SYMSOL are missing. The dataset is designed to assess performance on symmetric objects, so methods that predict distributions over SO(3) are necessary; the baseline Image2Sphere handles it well. I believe the proposed method might perform poorly on it because of the way the ground truth is computed or the MSE loss, which are the differences with respect to Image2Sphere. I believe a comparison and discussion of this possible limitation is warranted.

W4) The paragraph L113 "Rotation representation in frequency domain." is confusing. It talks generally of 3D rotations but actually describes the specific case of rotations of spherical functions. I think the U in eq (1) should be the same as the D in eq (2) but they are described as different things.

## References
[1] Klee et al, "Image to Sphere: Learning Equivariant Features for Efficient Pose Prediction", ICLR'23.

**Questions:**

In L212, is looks like S is a function on the sphere defined in the spatial domain, but it is said to be C'xN where N is the number of spherical harmonics. Should N be the number of points on the sphere instead?

Typos:

L44: leverages -> leverage

L70: faciltate -> facilitate

L87: Foruier -> Fourier

**Limitations:**

Possible limitation on symmetric objects should be discussed (see W3).

---

> ### Author Rebuttal · Authors · 2024-08-07
>
> We thank reviewer HCET for constructive comments and suggestions.
>
> **[W1. Clarifying our contributions compared to I2S [34]]**
>
> Please see our general response above.
>
> **[W2-a. Conversion from Euler angles to Wigner-D during training (L245)]**
>
> The conversion from Euler angles to Wigner-D matrices involves a mathematical transformation using the $ZYZ$ sequence of rotations and the corresponding Wigner-D matrix elements. The model outputs are indeed the coefficients associated with the Wigner-D matrices. The loss function in Eq. (5) compares these predicted coefficients directly with the ground-truth coefficients, facilitating the direct regression of 3D rotations in the frequency domain.
>
> Additionally, unlike Image2Sphere [34], we do not perform the inverse Fourier transformation (iFT) of the output of the networks during training. Instead, we train on the output Wigner-D coefficients in the frequency domain, and we do not apply the iFT even at test time.
>
> We hope this explanation clarifies the process and resolves any confusion. Thank you again for your valuable feedback.
>
>
> **[W2-b. Computing the similarity with the SO(3) grid during inference (L263)]**
>
> The similarity calculation in L263 is different from an inverse SO(3) Fourier transform. It is a process to find candidate Euler angles from the predicted Wigner-D coefficients. The predefined SO(3) HEALPix grid includes a set of predefined Wigner-D coefficients corresponding 1:1 to Euler angles. We calculate the similarity between this grid and the predicted Wigner-D coefficients to obtain a discretized distribution. This probability distribution provides multiple hypotheses for the pose (i.e., Euler angles) predictions of the input image. We then use argmax or gradient ascent to determine the final predicted Euler angles at inference, as shown in the results of Table R5. This method is similar to the approaches used in [48, 34].
>
> While an inverse SO(3) Fourier transform could convert the predicted Wigner-D coefficients to Euler angles, it has high time complexity, as it requires separate calculations for each frequency level. The SO(3) iFT method produces a single value, making distribution modeling difficult.
>
> In contrast, our design choice of querying the predefined HEALPix SO(3) grid enables distribution modeling by calculating simple vector-matrix multiplication. This allows us effective symmetric object modeling by combining the existing cross-entropy loss [48, 34], as demonstrated in Table A1 and Figure A1.
>
> **[W3, L1. Evaluation on SYMSOL]**
>
> Please see our general response above.
>
> **[W4. Clarification on paragraph L113 "Rotation representation in frequency domain"]**
>
> The paragraph "Rotation representation in frequency domain" describes how 3D rotations are represented in the frequency domain using spherical harmonics and the Wigner-D matrix.
>
> The Wigner-D rotation representation in L113-125 is not limited to a specific case of 3D rotations but can be converted from any 3D rotation representation, such as Euler angles, quaternions, and 3D rotation matrices. Our SO(3) equivariant network predicts the Wigner-D representation in the frequency domain instead of predicting rotations in the spatial domain (Euler angles, quaternions, etc.).
>
> To address the confusion, we clarify that the $U$ in Equation (1) and the $D$ in Equation (2) are indeed the same. We will update the manuscript to use consistent notation in Equations (1) and (2). We apologize for any confusion caused by this inconsistency.
>
>
>
> **[Q1. Clarification of the spherical notation in L212]**
>
> In Section 4.3, $\mathcal{S}$ is a spherical harmonics function defined in the frequency domain.
> $N$ denotes the total number of spherical harmonics in the frequency domain.
> $p$ denotes the number of points on the sphere in the spatial domain.
> We will update the manuscript to clearly indicate that  $\mathcal{S}$ is defined in the frequency domain and ensure the distinction between $N$ (number of spherical harmonics) and $p$ (number of points on the sphere) is clear.
>
>
> **[Q2. Typos]**
>
> We will revise our manuscript according to your findings of typos. Thank you for your feedback.

---

> > ### Comment · Reviewer_HCET · 2024-08-12
> >
> > W1: Thank you, I really hope this is updated in the main text since multiple reviewers pointed out that the method section might be seen as claiming contributions from prior work.
> >
> > W2-a: I am interested precisely in the "mathematical transformation" mentioned in the rebuttal which does not seem to be described anywhere. Specifically, the Euler angles represent an element of SO(3), while each Wigner-D matrix element is a fixed map from SO(3) to $\mathbb{C}$. So I think the map is from an element of SO(3) to a set of coefficients corresponding to the Wigner-D matrix elements; one way to obtain this map is by the inverse SO(3) Fourier transform of the impulse function on SO(3) centered on the ground truth rotation, but this procedure is not described anywhere so I am not sure if that's what is being done or I am missing something.
> >
> > W2-b: "The predefined SO(3) HEALPix grid includes a set of predefined Wigner-D coefficients corresponding 1:1 to Euler angles" -> sounds like this is exactly what the inverse SO(3) Fourier transform of function that is one at the cell corresponding to some Euler angles and zero elsewhere would give. So my understanding is that instead of computing the IFT of the predicted coefficients, the IFTs are precomputed for each discrete delta function of the grid, and the similarity between predicted coefficients and the precomputed is used as a distribution on SO(3). So I believe that for functions of bandwidth $B$ the approach stores $O(B^{6})$ coefficients (the grid has $O(B^{3})$ points and each delta function is represented by $O(B^{3})$ coefficients). The procedure to compute the distribution would also be $O(B^{6})$, which is the same as the naive algorithms for the SO(3) Fourier transform (SOFT). Since faster algorithms can reduce that to $O(B^{4})$ and even $O(B^{3}\log^{2}(B))$, I think the proposed method is actually slower and uses significantly more memory than just computing the IFT of the predicted coefficients. Please clarify.
> >
> > "The SO(3) iFT method produces a single value, making distribution modeling difficult." -> This seems incorrect; Fourier transforms and inverses are maps from function to function, not single values.
> >
> > W3: Thank you for adding SYMSOL experiments. They show that the proposed MSE and the previously used likelihood losses are complementary and combining both may help -- I think this is a much more convincing result than the current ones in the submission. It should be noted however, that it still doesn't surpass IPDF results on SYMSOL and I believe there are follow-ups that outperform it.

---

> ### Author Response · Authors · 2024-08-13
> **Response of the Official Comment by Reviewer HCET**
>
> Thank you for your thorough comment and insightful feedback.
>
> **[Reply to W1]**
>
> We will ensure that the main text is updated to reflect the clarifications provided in the rebuttal, as multiple reviewers have raised concerns about the clarity of the method section.
>
>
> * * *
> **[Reply to W2-a]**
>
> We apologize for any lack of clarity in our original submission regarding the "mathematical transformation”. Specifically, we map the Euler angles, which represent an element of $ SO(3) $ , to a set of coefficients corresponding to the Wigner-D matrix elements. These coefficients are critical in capturing the rotational properties in the frequency domain.
>
> **Mapping from SO(3) to Wigner-D Coefficients:**
> As you correctly pointed out, each Wigner-D matrix element provides a fixed map from $ SO(3) $ to $ \mathbb{C} $. Our method involves predicting these Wigner-D coefficients directly, bypassing the complexities and potential pitfalls of spatial domain parametrizations.
>
> To achieve this, we utilize an equivariant network to predict the Wigner-D matrix coefficients in the frequency domain directly from the input image features. The prediction process does not explicitly involve an inverse $ SO(3) $ Fourier transform of an impulse function centered on the ground truth rotation. Instead, our network learns to map the input features to the Wigner-D coefficients that represent the corresponding $ SO(3) $ rotation, leveraging the equivariant properties of the network to ensure rotational consistency.
>
> The output of our network is a vector of Wigner-D coefficients that directly encode the rotation in the frequency domain. These coefficients are then converted back to a rotation matrix or another suitable representation (e.g., Euler angles) as needed for evaluation or further processing.
>
> **Clarification of the GT Transformation:**
> We would like to gently remind you that Appendix A (and B.2) contains the detailed conversion equations between the Euler angles and the Wigner-D matrix, including the small Wigner-d matrix. We will further revise the transformation process in our final manuscript, elaborating on how the conversion between Euler angles and Wigner-D matrices is handled in our approach.
>
> * * *
> **[Reply to W2-b]**
>
> Thank you for your detailed feedback on the computational efficiency of our SO(3) grid generation and similarity computation for SO(3) distributions. We would like to clarify and justify our approach.
>
> **Clarification on the Inference Procedure and Similarity Calculation**:
> The similarity calculation described in L263 is designed to identify candidate Euler angles from the predicted Wigner-D coefficients by comparing them against a predefined SO(3) HEALPix grid at inference.
>
> You are correct that our approach involves precomputing the inverse SO(3) Fourier transforms to generate the set of predefined Wigner-D coefficients. This approach indeed requires significant space complexity, with a high time complexity of $ O(B^6) $ naively to initialize the predefined SO(3) grid. Consequently, our inference method is actually slower and uses more  significantly more memory than just computing the IFT of the predicted coefficients.
>
> However, after the predefined SO(3) grid is initialized, both the training and testing phases benefit from this precomputation, leading to reduced computation times during these phases. Additionally, for repeated runs, the grid can be stored and reloaded as needed, further optimizing execution time.
>
> We apologize for the inaccurate expression: "The SO(3) iFT method produces a single value, making distribution modeling difficult”. What we intended to convey was that the inverse SO(3) Fourier transform results in a specific mapping, not a single value.
>
> We hope this explanation addresses your concerns and clarifies our design choices. Our approach aims to balance computational feasibility with the ability to accurately model complex pose distributions in SO(3) space.
>
> * * *
> **[Reply to W3]**
>
> Thank you for acknowledging the addition of the SYMSOL experiments. As you noted, the proposed MSE loss and the previously used likelihood loss work complementarily. However, our model does not surpass the performance of I-PDF [48] on SYMSOL, and we agree that future research should focus on further improving symmetry modeling to address these challenges.

---

> > ### Comment · Reviewer_HCET · 2024-08-14
> >
> > ## Mapping from SO(3) to Wigner-D Coefficients:
> > Unfortunately it is still not clear to me how the map from Euler angles to Wigner-D coefficients happen. The appendices A and B mostly repeat textbook information that is not helpful:
> >
> > - definition of spherical harmonics (6)
> > - relation of wigner D and d (7)
> > - rotating spherical harmonics with Wigner-D (8)
> > - rotation in 3D from Euler angles (A.2.1)
> > - expression for Wigner-D (9)
> > - different ways to represent rotation (B.1)
> > - rotation of spherical harmonics again (10) -- same as (8).
> > - relation of wigner D and d again (11) -- same as (7).
> > - expansion into spherical harmonics (12).
> > - rotation of spherical harmonics coefficients (13).
> > - decomposition into spherical harmonics (14).
> >
> > Another suggestion is to clean up the repetitive writing. For example equations (1), (8) and (10) are the same and show the rotation of the spherical harmonics using Wigner-Ds which doesn't even seem relevant to the approach. Equations (2), (7) and (11) are also the same.
> >
> > Another guess at what might be happening is that the Wigner-Ds might be *evaluated* at the ground truth Euler angles to produce the supervision signal so the model is trained to predict a set of *values* of Wigner-D functions and not the Wigner-D coefficients that come out of a SO(3) FT. This would look a lot like "positional encoding" that maps coordinates to their evaluation in Fourier basis (sin/cos for the Euclidean spaces). But then I don't see how the output Wigner-D coefficients of the spherical CNN are mapped to the evaluation of the same coefficients at the ground truth. This would also contradict most of the text that refers to "Wigner-D coefficients" and not Wigner-D evaluations.
> >
> > ## Clarification on the Inference Procedure and Similarity Calculation:
> > I think computing the similarity against $O(B^{3})$ vectors of $O(B^{3})$ coefficients each is also $O(B^{6})$ even if all vectors are precomputed so doing a fast SOFT would still be faster?
> >
> > ## Conclusion
> > I think significant rewrite is needed, both to clarify technical details and also the contributions wrt Image2Sphere [see W1] so I do not recommend acceptance at this time.

---

### Official Review · Reviewer_HnJh · 2024-07-09

**Soundness:** 3
**Presentation:** 3
**Contribution:** 2
**Rating:** 6
**Confidence:** 4

**Summary:**

The paper proposes a method for regressing the rotation of an object from an image, where several (~10s) of training views of the object with known rotations are available. In particular, the approach proposes a rotation-equivariant network that predicts continuous Wigner-D matrix coefficients in the frequency domain, which can be converted into a discretised heatmap in SO(3) to extract the predicted rotation(s). The contributions are (1) a method for predicting in the SO(3) frequency domain directly and (2) an approach for decoding this to a rotation at inference time. The approach is evaluated on two datasets with respect to rotation error and demonstrates good performance. Ablations and sensitivity analyses in the paper and appendix validate many of the design choices.

**Strengths:**

S1. Originality. The idea of directly predicting Wigner-D matrix coefficients is original and is a sound design decision. It avoids some of the discretisation limitations of prior work and is intuitive.

S2. Quality. On the whole, the paper is well-presented, with useful figures and a clear structure. The writing is also quite decent, and the results are clearly displayed. In particular, the related work is well-written, clear, concise and sufficiently complete to position the work in its research context. In addition, the experiments validated the overall efficacy of the approach well, except for a couple of points below, especially the graphs of performance w.r.t. training data cardinality.

S3. Clarity. The paper is clear overall and does a good job of directing the reader's attention appropriately. One caveat is that it is not self-contained (with respect to the main paper and also the main+appendix), with some components left out of the main treatment that reduced the overall clarity of the treatment. Nonetheless, the introduction motivated the problem and approach very well, and each section was itself well-motivated, making the paper easy to follow.

S4. Significance. The paper's contributions of predicting the frequency-domain coefficients and decoding them to a heatmap are likely to be of positive but mild significance to the NeurIPS community. The task itself is of high significance, since rotation estimation from an image (for known objects) is a challenging task that is important for many applications, including factory robotics, augmented reality, automotive, etc.

**Weaknesses:**

W1. Contributions/claims.
W1.1 The method in sections 4.1-4 is presented as part of the contribution, but appears to be the same as previous work, including [34]. If this is not the case, the design differences and the rationale should be better explained. If it is the case, this limits the contribution to sections 4.5-6, which is (a) directly predict and optimise the Wigner-D matrix coefficients, and (b) convert to a discretised heatmap and take the argmax. While (a) is likely key to the success of the method, it would appear to be a small variation on an existing approach; (b) on the other hand is very minor and indeed not claimed as a contribution. What this reviewer would like to see here is an expansion of these sections with some interrogation (and testing, in section 5) of the design choices. For example, is MSE a theoretically-grounded choice here, what are the weights w, why argmax instead of clustering for multi-hypothesis outputs?

W1.2 Relatedly, the list of contributions has 3 items, but 2-3 do not seem to be contributions of this paper. That is, the contribution of leveraging SO(3)-equivariant representations to guarantee equivariance has clearly been done before (e.g., [27,34,40,33], among others); and achieving SOTA performance is not a contribution.

W1.3 The claim that the proposed method is continuous and therefore better than the discretised methods (e.g., line 97) is dubious, when the predicted rotations are indeed discretised (section 4.6). The claim of loss of precision due to discretisation (for other methods) is also not validated in the experiment section; and one would assume that this method would also lose precision for the same reason (as Q decreases).

W1.4 The claim that parametric models are insufficiently expressive for this task (L82) is not validated. It seems likely, but would depend on the model and would ideally be directly tested to back up the claim.

W2. Design choices. (Partially overlaps with above)
W2.1 Spherical mapper. While this is taken from existing work, it is not explained why it's a reasonable design decision to warp a 2D planar signal to the sphere, or given any motivation. It's clear that it's necessary for the method, to allow S2- and SO(3)-equivariant operations, but geometrically it seems quite dubious. Could the feature warping be made more geometrically meaningful by first running the image through DepthAnythingv2 and projecting out from a point on the crop's centroid ray?

W2.2 The text mentions other approaches for non-linearities that avoid the extra (i)FFT steps (L233) but does not test or validate this design choice.

W2.3 MSE (5). This does not seem to be a theoretically- or empirically-justified choice but is key to the contribution. Lines 254-258 loosely refer to this, but this section would seem to be a good opportunity to provide a justification for the design.

W2.4 Discretised distribution on SO(3). The design choices around this part are not interrogated or justified - it reads as rushed or incomplete. For example, it states that the inference scheme "models objects with ambiguous orientations or symmetries by employing multiple hypotheses", but this is never done, as far as I can see. Instead, the argmax or gradient ascent is used, returning a single mode. Nowhere does the paper demonstrate how to use the approach to extract multiple hypotheses or test efficacy on extracting equivalent rotations for symmetric objects (see experiment section below). This is a missed opportunity and leads to the paper arguing against itself.

W3. Clarity.
W3.1 As alluded to earlier, the treatment is not self-contained. In particular, the Wigner-D matrix representation is central to the method and, for completeness and ease of understanding, should be included in the main paper. The reader would want to see the formulation that directly connects the task to the representation, instead of leaving that jump a bit vague (L121) and referring to the appendix. The function f in (4) and (12) are presumably not the same, but are not specifically defined for this task. Essentially, it reads as being incomplete, missing some of the material that connects the theory to the specific instantiation here.
W3.2 L248. Please provide/derive the weights w for completeness.

W4. Experiments.
W4.1 The comparisons (especially in Table 1) would be more meaningfully grouped by the backbone extractor. For example, we should actually be comparing the similar method [34] with "ours (ResNet-50)", where a 0.6deg improvement is achieved. Similarly, it would be useful to compare the ResNet101 version on [34] in this table (like the version in Tab 2) since it is the closest existing method.
W4.2 Missing dataset. SYMSOL, used in the closest related work [34,40], is not included in this paper, despite being a good opportunity to evaluate the performance with complex symmetries. Leaving it out gives the impression that the method cannot handle such cases, even though it seems that it ought to be able to.
W4.3 Multi-hypotheses untested (despite L271). This claim ought to be experimentally tested, since it is such a relevant and interesting potential feature of the approach.
W4.3 Ablations/sensitivity analyses. A sensitivity analysis on the influence of discretisation on precision (Q) is missing (despite claim in L96); the choice of MSE is not tested.


W5. Minor.
W5.1 The paper needs a proofread - there are many typos, syntactical and grammatical errors.
W5.2 Confusing formatting error with footnote 2 appearing the page before it appears in the caption (L262.5).

**Questions:**

1. Is 4.1-4 pre-existing work? If so, please make this clear in the paper.
2. Recommendation to rework the list of contributions to focus on the new aspects of the proposed approach and their significance.
3. Recommendation to validate the claim that the method does not lose precision due to discretisation.
4. Recommendation to expand on the design decisions W2.1-4.
5. Recommendation to test objects with complex symmetries via the SYMSOL dataset; and test the ability to predict multiple hypotheses as claimed.

**Limitations:**

Yes, the authors adequately address the limitations and broader impact in the appendix, with an appropriate level of self-reflection and consideration.

---

> ### Author Rebuttal · Authors · 2024-08-07
>
> We thank reviewer HnJh for constructive comments and suggestions.
>
> **[W1.1, W1.2, Q1, Q2 clarifying the contributions]**
>
> Please see our general response above.
>
> **[W1.3, W4.4, Q3. Continuity of rotations, Sensitivity analysis of the SO(3) HEALPix discretization]**
>
> We carefully claim that our learning method focuses on continuous rotations. We directly learn the Wigner-D coefficients, which are derived from 3D rotations (Euler angles), without any discretization during the training phase. The use of the SO(3) HEALPix grid during inference serves two purposes:
>
> 1. To convert SO(3) rotations from the frequency domain to the spatial domain, and
> 2. To address pose ambiguity by providing multiple solutions.
>
> As a result, we obtain a distribution with very sharp modality. By taking the argmax of this distribution, we achieve sufficient precision in 3D orientation estimation, specifically around 1.5 degrees.
>
> Table R2 shows the effect of changing $Q$ on performance is. For the ModelNet10 benchmark, we achieve similar results in the common target metrics such as Acc@15 and Acc@30 even with a lower size of SO(3) discretization grid ($Q$=4.6K).
> With our model's choice of $Q$=2.36M, we obtain high scores even on low-threshold evaluation metrics like Acc@3. Comparative experiments at lower thresholds are provided in appendix Table A1.
>
> Therefore, the continuity of rotations in our method ensures that we can predict an accurate, more precise pose. This is a significant advantage over other methods that suffer from precision loss due to discretization, as our approach maintains high accuracy across varying levels of discretization.
>
>
> **[W1.4 The claims of parametric model]**
>
> Our statement about the insufficient expressivity of parametric models (L82) refers to the potential lack of flexibility due to the dependency on predefined prior models, compared to non-parametric models. We acknowledge that this claim was not directly validated in our work. To provide a more accurate representation, we will revise our claim to reflect that.
>
> **[W2.1. Justification of the spherical mapper]**
>
> The spherical mapper maintains the geometric structure of the image when projecting onto the S2 sphere, as detailed in [34]. This method involves lifting the 2D image onto the sphere and converting spherical points using spherical harmonics. Table R6 shows that the spherical mapper outperforms simple Fourier transforms on 2D feature maps.
>
> Using depth information from methods like DepthAnythingv2 for 3D lifting is a good idea and can enhance geometric accuracy. Additionally, centroid ray regression has been explored in research such as [70]. However, incorporating external depth modules increases computational costs and broadens our research scope, so we consider this for future work.
>
>
> **[W2.2 Non-linearites for equivariant layers]**
>
> The FFT-based approximate non-linearity [19] and equivariant non-linearity for tensor field networks [51,65] mentioned in the text provide non-linearity based on Fourier kernels, avoiding the need for FFT and iFFT steps. However, these methods are not included in our quantitative evaluation as their code is not publicly available. Testing these approaches could potentially improve performance or efficiency when combined with our Fourier-based SO(3) equivariant network.
>
> **[W2.3, W.4.3, Q4 Design choice of MSE]**
>
> Please see general response above.
>
> **[W.2.4, W4.2, W4.3, Q5. Evaluation on SYMSOL]**
>
> Please see general response above.
>
> **[W2.4 Discretised distribution on SO(3). (argmax vs. clustering)]**
>
> Table R5 shows the evaluation results using gradient ascent on the pose distribution.  While gradient ascent does provide some performance improvement, the increase in inference time outweighs these gains, so argmax is our preferred method for simplicity and fast evaluation.
>
> **[W3.1 Clarification of Wigner-D representation, ]**
>
> We will clarify the Wigner-D matrix representation in the main paper. While detailed equations are provided in Sections A.2, A.2.2, and B.2 of the appendix, we will include a concise explanation in the main text.
>
> Specifically, the Wigner-D representation is implemented in a flattened form at different frequency levels. For example, the matrix coefficients in a frequency level $l$ is represented as a flattened vector of size $(2l+1)*(2l+1)$. We use a maximum frequency level of 6, resulting in a vector of size 455 (i.e., $1∗1+3∗3+5∗5+...+13∗13=455$) for the Wigner-D coefficients. This will be clearly explained to directly connect the task to the representation in our final manuscript.
>
> **[W3.1 Notation of $f$ in eq (4) and (12)]**
>
> We will clarify the notation to differentiate the two $f$ functions in Equations (4) and (12). These functions are indeed different.
> Equation (4) represents the general form of a group equivariant network.
> Equation (12) describes the rotation of spherical harmonics using the rotated coefficients of the Wigner-D matrix.
> We will update the notation to clearly distinguish between these two functions and ensure the connection between the theory and its specific instantiation is clearly defined.
>
> **[W3-2. Derive the weights $w$]**
>
> Please see our response above.
>
> **[W4.1. Fair comparison of  the backbones in Table 1]**
>
> Thank you for the suggestion. We include a comparison in the appendix (Table A1) that aligns the sizes of the backbone networks. Some networks, like Inception-v3, are not directly comparable, so we group representative methods for fair comparison based on similar backbones. These results show that our method outperforms existing methods, even on finer metrics like Acc@3 and Acc@5, demonstrating its effectiveness.
>
>
> **[W5. Typos and formatting errors]**
>
> We will conduct a thorough proofreading to correct typos, syntactical, and grammatical errors. Additionally, we will adjust the positioning of footnote 2 to ensure it appears at the correct location where it is cited in the text. Thank you for your helpful suggestions.

---

> > ### Comment · Reviewer_HnJh · 2024-08-12
> >
> > Thank you for your detailed response, and in particular the results included in the PDF. My primary concerns---regarding (W1) the contribution overlap with [34], (W2) the design choice of the MSE, and (W4) the lack of evaluation on SYMSOL and its symmetric objects---have largely been addressed in the rebuttal.
> >
> > Specifically, the authors have committed to moving the material from existing work into a background section, making it substantially clearer what contributions have been made. While the contribution is relatively small in my view, it is sufficient and the material in the rebuttal has strengthened it to an extent (Table R1 especially). The authors have also interrogated the choice of MSE quite convincingly, addressing its strengths and shortcomings, and ablating it in Table R4. Finally, and most importantly for my decision, the authors have evaluated on the SYMSOL dataset, showing that the method performs poorly for highly-symmetric objects, but interestingly can be combined with the loss from [34] to achieve greater performance than either, on this dataset.
> >
> > A small note in relation to the rebuttal: I maintain that the claim that the proposed method is continuous and therefore better than the discretised methods (e.g., line 97) should be removed. I understand that it is continuous in the training loop, but it most certainly suffers from discretisation at inference time. This claim should be more carefully worded, especially now that the authors have good evidence regarding this effect (in Table R2).
> >
> > Having carefully read the rebuttal and the responses from and to the other reviewers, I am inclined to increase my rating to a WA.

---

> > > ### Author Response · Authors · 2024-08-13
> > > **Response of the Official Comment by Reviewer HnJh**
> > >
> > > Thank you for increasing the score to WA and acknowledging that our rebuttal successfully addressed concerns W1, W2, and W4.
> > >
> > > We will remove the claim that the proposed method is continuous and therefore better than discretized methods, as in L97, by following your suggestion. We recognize that while the method is continuous during training, it does undergo discretization during inference. Consequently, we will carefully rephrase this point to emphasize the continuity of our method during training.

---

### Official Review · Reviewer_iMp8 · 2024-07-13

**Soundness:** 3
**Presentation:** 2
**Contribution:** 3
**Rating:** 5
**Confidence:** 3

**Summary:**

In this work authors predicts SO(3) poses for objects by predicting Wigner-D coefficients in frequency space. Similar to other work [1], it first lifts 2D features to 2-sphere using pre-defined grid and orthographic projection to sphere using this grid, convert them to frequency domain, applies SO(3)-equivariant spherical convolution on top to predict a vector of Wigner-D coefficients. At inference time, these Wigner-D coefficients are mapped back to spatial rotation R.

Authors show experiments on multiple datasets showing better accuracy compared to other work. Authors also show few-shot views training accuracy and ablation study for different design components of the network.


References

[1] Klee, David M., et al. "Image to sphere: Learning equivariant features for efficient pose prediction." arXiv preprint arXiv:2302.13926 (2023).

**Strengths:**

Overall, the whole paper is well organized with sufficient figures to aid in understanding. Most of the math needed is included in the paper or cited the right resources. Experiments show that method is outperforming other pose-estimation methods that works in spatial / frequency domain. Ablation studies confirm authors claim of different network components.

Authors also provide sufficient detail on the model architecture, spherical convolution design, and training hyper-paramters, suggesting reproducibility.

Appendix has lots of nice extra experiments along with inference time.

**Weaknesses:**

Some pose visualization on objects would be nicer and should be compared to other methods.

**Questions:**

How would this method work for in-the-wild datasets?

Would this method be able to handle occlusions?

If we have to use "transformers" instead of convolution, what would change?

**Limitations:**

Authors discuss limitations of their approach in the last section. Societal impact is discussed in appendix in detail.

---

> ### Author Rebuttal · Authors · 2024-08-07
>
> We thank reviewer iMp8 for constructive comments and suggestions.
>
> **[W1. Comparison of pose visualization]**
>
> We present a comparison of pose visualizations in Figure R2. This visualization method is the same to those used in Figures A2 and A3 in the appendix.
>
> We compare to the I2S [34] baseline. The left side of Figure R2 shows results from ModelNet10-SO(3), while the right side presents on PASCAL3D+. The numbers next to "Err" above the input images represent the error in degrees between the model's predicted pose and the ground truth (GT) pose.
>
> These results demonstrate that our model provides more accurate and precise pose estimations, even in cases where the I2S baseline fails. Additionally, on the PASCAL3D+ benchmark, which includes objects captured in real-world scenarios, our model consistently shows correct pose estimations, particularly in challenging situations where the I2S baseline struggles.
>
> **[Q1. Evaluation on in-the-wild datasets]**
>
> The PASCAL3D+ dataset is an in-the-wild dataset captured in typical real-world environments, which we evaluated in Table 2 and Figure R2. As far as we know, PASCAL3D+ is the primary in-the-wild dataset used for 3D orientation estimation. We have not found any other in-the-wild datasets that are commonly used for this task.
>
>
>
> **[Q2. Handling occlusion]**
>
> Yes, our method is capable of handling occlusions.
>
> While our SO(3) equivariant estimator can offer some robustness to occlusions due to its rotational invariance and localized feature extraction, handling occlusions effectively often requires additional techniques and strategies. Our network can extract local features in a manner that is robust to 3D rotations, allowing it to identify visible parts of objects even when other parts are occluded. Additionally, the network's ability to recognize objects regardless of their orientation helps in identifying partially visible objects.
>
> The results in Table R1 for the SYMSOL II dataset, which includes self-occlusion scenarios in symmetric solids, demonstrate that
> our model jointly trained with  $\mathcal{L}_{\text{dist}}$ [48, 34] handle occlusions better than the I2S [34] baseline. Our model improves accuracy by directly predicting a single clear pose, unlike the baseline model.
>
> Figure R1 shows the SYMSOL II visualization results, specifically in examples 4, 5, and 6. When provided with occluded marks of symmetric objects, our model effectively uses these cues to estimate a single pose.
>
> **[Q3. Transformer instead of convolution]**
>
> We choose a convolution-based design, because using transformers instead of convolutions results in a slight performance drop.
>
> Table R3 presents the results when transformers are used as the backbone network instead of convolutions. We trained the model using a Vision Transformer (ViT) pre-trained backbone pre-trained by the geometric task of cross-view masked image modeling [A].
>
> Although the ViT is heavier and requires longer training time (1.4x), its performance actually declines. This suggests that convolutional backbones may still be more effective for 3D orientation estimation tasks.
>
> [A] CroCo: Self-Supervised Pre-training for 3D Vision Tasks by Cross-View Completion (Weinzaepfel et al., NeurIPS 2022)

---

> > ### Comment · Reviewer_iMp8 · 2024-08-09
> > **Thanks for the rebuttal.**
> >
> > Reply W1: I was rather asking to fit bounding boxes onto the object to compare with Ground-truth. Errors on sphere doesn't show exactly, how well the predicted pose is fitting on the object.
> >
> > Reply Q1: I meant the performance on dataset outside the training distribution. Does the model generalize to different dataset that the one it is trained on?

---

> ### Author Response · Authors · 2024-08-11
> **Response of the Official Comment by Reviewer iMp8**
>
> First, thank you for taking the time to read and consider our rebuttal.
>
> * * *
> **Reply to R1 regarding pose visualization comparison:**
>
> We would like to clarify that the visualization of pose distribution using the surface of the 2-sphere with the Mollweide projection [A] conveys the same information as visualizations that use bounding boxes or the $x$, $y$, $z$ axes. For example, the pose distributions shown on the spheres in Figures R2, A2, and A3 can be converted into single pose values that can be directly plotted onto the object, as demonstrated in Figure 6 of [B].
>
> [A] Implicit-PDF: Non-Parametric Representation of Probability Distributions on the Rotation Manifold (Murphy et al., ICML 2021)
> [B] A Laplace-inspired Distribution on SO(3) for Probabilistic Rotation Estimation (Yin et al., ICLR 2023)
>
> However, we believe that the visualization method on the 2-sphere using the Mollweide projection [A], as used in Figures R2, A2, and A3, offers the additional benefit of enabling multi-hypothesis plotting in 3D pose space. This allows for deeper insights into symmetry and pose ambiguity modeling.
>
> * * *
> **Reply to Q1 regarding in-the-wild datasets (OOD scenarios):**
>
> Evaluating out-of-distribution (OOD) performance is generally not the primary focus in the context of this 3D orientation estimation task. However, we have conducted OOD generalization experiments using our proposed method by training the model on different datasets, between ModelNet10 and PASCAL3D+. The results are presented below:
>
> | Training Dataset | Evaluation Dataset | Acc@15  | Acc@30  | Rot. Err. |
> |------------------|--------------------|---------|---------|-----------|
> | ModelNet-SO(3)   | ModelNet-SO(3)      | 0.7590   | 0.7668   | 15.08°     |
> | ModelNet-SO(3)   | PASCAL3D+           | 0.0004  | 0.0019  | 112.98°   |
> | PASCAL3D+        | ModelNet-SO(3)      | 0.0015  | 0.0086  | 130.44°   |
> | PASCAL3D+        | PASCAL3D+           | 0.7495  | 0.8965  | 8.92°     |
>
> **Table:** Cross-dataset evaluation for validating out-of-distribution generalization on ModelNet10-SO(3) and PASCAL3D+ datasets.
>
> As the results indicate, the model does not perform well when evaluated on out-of-distribution dataset. Nevertheless, we recognize this as an important area for future research, and we appreciate your suggestion to explore this further.

---

> > ### Comment · Reviewer_iMp8 · 2024-08-12
> >
> > I thank authors for the clarifications. I acknowledge the response.

---

### Author Rebuttal · Authors · 2024-08-07

We appreciate the reviewers for their constructive comments and recognition of the strengths of our paper:
* Reviewer iMp8: We appreciate your positive feedback on the paper's structure, figures, equations, detailed method section, and ablation studies.
* Reviewer HnJh: Thank you for highlighting the strengths of our Wigner-D prediction in overcoming discretization method limitations. We value your compliments on the clarity and structure of our manuscript.
* Reviewer HCET: We appreciate your recognition of our use of a loss function in the spectral domain and the superior performance on benchmarks.
* Reviewer f5v8: Thank you for acknowledging our novel approach to object pose estimation and the thoroughness of our experimental section.

We address common questions in below:

##  1. Contributions

**R: HnJh [W1.1, W1.2, Q1, Q2. Clarifying the contributions, List of contributions]**

**R: HCET [W1. Clarifying our contributions compared to I2S [34]]**

**R: f5v8 [W1. Clarifying contributions]**


We would like to clarify that while our method shares the same backbone structure as [34], the key innovation lies in how we optimize using the Wigner-D matrix directly. This is a significant departure from [34], which does not utilize the Wigner-D matrix in this manner. Specifically, we propose a continuous rotation learning method through direct Wigner-D prediction.

We clarify that Sections 4.1-4 describe foundational concepts and methodologies based on pre-existing work, which may give the impression that these are new contributions. To address this, we will revise the paper to make it clear that these sections are based on prior work and move them to the background section. This will better highlight our novel contributions in Sections 4.5-6.

To address the reviewer's concerns, we clarify and emphasize the novel aspects of our contributions:
1. *Frequency-Domain Prediction*: Our approach uniquely predicts Wigner-D coefficients directly in the frequency domain, avoiding issues like discontinuities and singularities in traditional spatial domain methods, ensuring precise and continuous pose estimation.
2. *Tailored MSE Loss*: We introduce a frequency-domain specific Mean Squared Error (MSE) loss. This tailored loss function supports continuous training for SO(3) pose estimation and has the potential to integrate cross-entropy loss for distribution modeling, effectively addressing object symmetry challenges.
3. *Superior Performance and Data Efficiency*: Our SO(3)-equivariant network consistently outperforms existing methods on standard pose estimation benchmarks, demonstrating data sampling efficiency in data-limited scenarios.

These contributions provide substantial improvements in accuracy and robustness over the baseline methods, as demonstrated in our experimental results. We believe that our approach represents a meaningful advancement in the field of 3D orientation estimation.


## 2. Results on SYMSOL


**R: HnJh [W.2.4, W4.2, W4.3, Q5. Evaluation on SYMSOL, Test of the distribution on SO(3).]**

**R: HCET [W3, L1. Evaluation on SYMSOL]**

**R: f5v8 [W3-3. Effects on Wigner-D regression loss in symmetric objects]**

Table R1 shows symmetric object modeling on the SYMSOL datasets. Compared to the first row and second row [34], our model with only the Wigner-D regression loss derives on sharp modalities, which can be less effective than [34] for symmetric objects in SYMSOL I.

For clearly defined pose cases (e.g., SphereX in SYMSOL II), our Wigner-D loss alone performs well. However, in other SYMSOL II scenarios, the sharp distributions produced by our model can lead to low average log likelihood scores. This metric is particularly harsh on models with sharp peaks, making them vulnerable to very low scores in some failure cases.

In the third row, joint training of our method with the distribution loss [48, 34] achieves better performance than the baseline [34], demonstrating its ability to model symmetric objects. These results highlight our method's potential in handling complex symmetries and predicting multiple hypotheses. Figure R1 shows the visualization of pose distribution.

Most real-world objects have unique, unambiguous poses, validating our single pose regression method (e.g., ModelNet10-SO3, PASCAL3D+). If the task needs to cover symmetric cases, our model can be modeled with distribution loss [48, 34].


## 3. Design choice of MSE


**R: HnJh [W1.1, W2.3, W4.3, Q4. Design choice of MSE]**

**R: f5v8 [W3-2. Comparison with other loss functions]**

The choice of MSE for regression loss is for its simplicity and effectiveness. Cosine and angular distances measure only angles between vectors, ignoring magnitude, which is crucial in frequency domain applications. Chordal and geodesic distances often require normalization to the unit sphere, making them less intuitive and more computationally intensive, which can be a drawback in practical terms.

Table R4 compares various loss functions. While Huber and L1 losses are alternatives, they do not perform as well as MSE in our context. Geodesic loss in the frequency domain is ineffective because it requires separate calculations for each frequency level of the Wigner-D matrix, potentially losing the precision of the original 3D rotation, as we truncate the Fourier basis at a frequency level of 6.

**R: HnJh [W3-2. Derive the weights $w$ (L248)]**

**R: f5v8 [W3-1. The value of $w_l$]**

Figure R3 shows visualization of the weights $w$. The figure shows a consistent learning pattern, even as the # of training views changes. Notably, the weights increase with higher frequency levels, indicating that more complex rotations are effectively modeled with greater emphasis on higher frequencies. This demonstrates the effectiveness of weighting more complex rotations more heavily during training.

We have expanded these sections and included the relevant experiments in Section 5 to provide a comprehensive understanding of our design choices.

---

### Decision · Program_Chairs · 2024-09-25

**Decision:**

Accept (poster)

**Comment:**

Initially there was some confusion about the similarity to [34] and the contribution. The authors promised to update the paper according to the rebuttal. As reviewer HnJh states "While the contribution is relatively small in my view, it is sufficient and the material in the rebuttal has strengthened it". Three reviewers (HnJh, f5v8, iMp8) agree on this statement.

The fourth reviewer (HCET) has reservations because many things were not clear initially and need to be udpated in the manuscript.

Overall, the AC recommends trusting the authors that they will update the manuscript according to the comprehensive discussion in the rebuttal and recommends to accept the paper.